# ASC specks as a single-molecule fluid biomarker of inflammation in neurodegenerative diseases

Evgeniia Lobanova [1,2] ✉, Yu P. Zhang[1,2], Derya Emin[1,2], Jack Brelstaff[3], Lakmini Kahanawita[3], Maura Malpetti [3], Annelies Quaegebeur[3], Kathy Triantafilou [4], Martha Triantafilou [4], Henrik Zetterberg [5,6,7,8,9,10], James B. Rowe [3,11], Caroline H. Williams-Gray [3], Clare Elizabeth Bryant [12] & David Klenerman [1,2] ✉

Immunotherapeutic strategies for Alzheimer's and Parkinson's disease would be facilitated by better measures of inflammation. Here we established an ultra-sensitive single-molecule pull-down immunoassay combined with direct stochastic optical reconstruction microscopy (dSTORM) to measure the number, size and shape of individual extracellular inflammasome ASC specks. We assayed human post-mortem brain, serum and cerebrospinal fluid of patients with Parkinson's and Alzheimer's as well as healthy elderly. The number of ASC specks increased and showed altered morphology in the blood of early-stage Parkinson's and Alzheimer's patients compared to controls, mimicking those found in the brain and cerebrospinal fluid. In serum samples we also measured the number of Aβ, p-tau and α-syn aggregates and formed a composite biomarker of (ASC + p-tau)/Aβ and (ASC + α-syn)/Aβ ratios that distinguished age-matched healthy controls from patients with early-stage Alzheimer's with AUC of 92% and early-stage Parkinson's with AUC of 97%. Our findings confirm ASC specks as a fluid candidate biomarker of inflammation for neurodegenerative diseases with blood being the main focus for further development as convenient sample for diagnostics and clinical trials.

Parkinson's disease and Alzheimer's disease (PD and AD) are the most prevalent neurodegenerative disorders, affecting 55 million people worldwide[1,2]. Although diagnosis of well-established disease can be accurate, based on clinical features and supportive diagnostic imaging, there is a pressing need for quantitative biological tools that work in early stage disease, allowing diagnostics and recruitment for early phase clinical trials[3]. There are disease-specific protein aggregates of α-synuclein (α-syn) in PD and amyloid-β (Aβ) and tau in AD, and both

[1]Yusuf Hamied Department of Chemistry, University of Cambridge, Lensfield Road, Cambridge CB2 1EW, UK. [2]UK Dementia Research Institute at University of Cambridge, Cambridge CB2 0XY, UK. [3]Department of Clinical Neurosciences, University of Cambridge and Cambridge University Hospitals NHS Trust, Cambridge, UK. [4]School of Medicine, Division of Infection and Immunity, University Hospital of Wales, Cardiff University, Cardiff, UK. [5]Department of Psychiatry and Neurochemistry, Institute of Neuroscience and Physiology, The Sahlgrenska Academy at the University of Gothenburg, Mölndal, Sweden. [6]Clinical Neurochemistry Laboratory, Sahlgrenska University Hospital, Mölndal, Sweden. [7]Department of Neurodegenerative Disease, UCL Institute of Neurology, Queen Square, London, UK. [8]UK Dementia Research Institute at UCL, London, UK. [9]Hong Kong Center for Neurodegenerative Diseases, Hong Kong, China. [10]Wisconsin Alzheimer's Disease Research Center, University of Wisconsin School of Medicine and Public Health, University of Wisconsin-Madison, Madison, WI, USA. [11]Medical Research Council Cognition and Brain Sciences Unit, University of Cambridge, Cambridge, UK. [12]Department of Medicine, Box 157, Level 5, Addenbrookes Hospital, University of Cambridge, Cambridge CB2 0QQ, UK. ✉e-mail: el519@cam.ac.uk; dk10012@cam.ac.uk

diseases are also characterised by chronic inflammation, which precedes and predicts clinical progression. The development of sensitive biomarkers of inflammation has potential to improve early diagnosis[4,5] particularly if used in conjunction with protein aggregation markers, and also to support stratification and monitoring in immunotherapeutic clinical trials. Inflammatory cytokines show low grade elevation in the blood in mild cognitive impairment (MCI)[6], AD[7] and PD[5] and may predict faster disease progression[8], but cannot distinguish reliably between the disease state and controls. However specific inflammatory molecules may be of more utility, such as the complement system component regulator, clusterin, which is elevated in prodromal AD ahead of conversion to clinically manifest disease[9]. Biomarker identification remains challenging due to a combination of extremely low concentration (e.g. plasma tumour necrosis factor (TNF)-α ~ 7 pg/mL[10], plasma Aβ42 ~ 20 pg/mL[11]) and heterogeneity in both size and structure of candidate protein biomarkers; many of which form aggregates in disease rather than monomers.

New methods are required which are sensitive and specific enough to identify the pathological species from the population of aggregates present in samples. ELISA and single-molecule array (Simoa) assays[12] have been developed and utilised for detecting picomolar concentrations of monomeric proteins in the blood and cerebrospinal fluid (CSF). However, these techniques do not distinguish aggregates over monomers or aggregates with different sizes and shapes. Super-resolution imaging of individual aggregates from the blood or CSF using direct stochastic optical reconstruction microscopy (dSTORM)[13] or DNA-PAINT[14] allows one to characterise in-detail the size and shape of single aggregates down to 30 nm resolution. This resolution, and the statistics of distribution of aggregate size and shape, provides additional metrics to distinguish healthy controls from people with early disease[15].

Aggregates of α-syn, Aβ and tau are sensed by the immune system via pattern recognition receptors (PRRs)[16–18]. These aggregates trigger a pro-inflammatory cascade through Toll-like receptors and NLRP3 inflammasome formation[19] in immune cells such as microglia, astrocytes and blood monocytes which, if not resolved, may lead to neuronal damage and ultimately neurodegeneration[20,21]. Inflammasome complexes contain an NLR (Nucleotide-binding domain, Leucine-rich Repeat-containing) protein, the adaptor protein Apoptosis-associated Speck-like protein containing a CARD (ASC) and an effector protein (caspase 1). Activation of inflammasomes results in cleavage of pro-IL-1β and pro-IL-18 (to form active inflammatory cytokines) and gasdermin D to trigger lytic cell death. During inflammasome activation ASC rapidly assembles into a large protein complex, up to 1 μm in size, termed the 'speck' which can, ultimately, be lost from the cell when it ruptures[22]. Detecting extracellular secreted ASC specks in human biofluids could provide useful information about disease mechanisms. Inflammasome activation contributes to AD and PD pathology in disease models[20,23]. There is increased expression of the NLRP3 inflammasome in human AD and PD brain and blood[24,25]. Hence, the detection of ASC specks in patient samples might be a useful marker of neuroinflammation that could help monitor disease progression when combined with measurement of α-syn, Aβ and tau aggregates.

We have established the single-molecule pull-down (SiMPull) detection assay as an optimal platform for highly sensitive and specific single-molecule imaging of various protein complexes present in brain, serum and saliva samples from neurodegenerative diseases[26–31]. In this study, we applied SiMPull for ultra-sensitive single-aggregate detection of inflammasome ASC specks in human biofluids. Combining the SiMPull assay with dSTORM, we measured the size down to 30 nm resolution, and quantified the shape and number of ASC aggregates present in biofluids. We tested this approach using biofluids, including serum, CSF and brain-derived samples, from people with early Parkinson's disease and Alzheimer's disease, compared to healthy aged controls. We also investigated whether the ASC speck has

a morphological phenotype specific for disease. We have previously used a similar method to detect a size difference between soluble β-sheet rich α-syn aggregates present in both CSF and serum samples from PD patients and controls as well as Aβ aggregates in CSF of AD patients in comparison to controls, with both having a higher proportion of larger aggregates (>150 nm) present in disease[14,15,29]. Combining ASC speck detection with analysis of Aβ, p-tau and α-syn aggregates using our SiMPull assay we explored the diagnostic accuracy of their different combinations (e.g.: *ASC/Aβ*, *p-tau + ASC*, *(p-tau + ASC)/Aβ*, *α-syn + ASC*, *(α-syn + ASC)/Aβ*, etc.). The aim of our study was to establish the utility of ASC as a sensitive and specific biomarker of disease state when combined with other biomarkers of protein aggregate diseases.

## Results
### Establishment of the ASC-SimPull assay for ASC specks detection in human biofluids
The workflow of the SiMPull method for detecting aggregated ASC in human biofluids is demonstrated in the Fig. 1A. Briefly, by immobilising single ASC aggregates/specks present in a sample with a biotinylated antibody against the protein on a glass surface, we selectively capture ASC. The same antibody labelled with a dye is added for single-molecule detection. Blocking the surface with polymer (polymer passivation) significantly reduces the fluorescence signal caused by non-specific antibody binding. If no sample is presented, the blocked surface will not bind any detection antibodies. By combining the SiMPull assay with dSTORM we can precisely measure the size, shape and number of ASC aggregates in biofluids and use this morphology information as additional metric to determine whether there are differences between healthy controls and people with PD and AD. Super-resolving ASC aggregates allows us to accurately distinguish between different sizes of ASC aggregates that are > 30 nm (our spatial resolution limit). The species that are < 30 nm will appear the same size and will be excluded during the data analysis allowing us to specifically quantify the activated inflammasome ASC complexes or specks rather than inactive monomers present in the sample.

We first proved that our method allows sensitive detection of the inflammatory response through measurement of ASC speck aggregates generated in THP-1 cells during inflammasome activation (Fig. 1B–F). The SiMPull assay is capable of detecting ASC specks in both the NLRP3 inflammasome-activated THP-1 cell lysates (LPS-primed nigericin-activated) and those secreted into the media compared to untreated negative controls (Fig. 1B). There was a gradual increase in the number of ASC specks detected in the THP-1 media when treated with LPS/nigericin over time with the maximum number of specks detected at 30 min nigericin (Fig. 1B). Significantly more specks were seen at 15 min nigericin incubation vs detection IgG Control (CTRL) media and capture/stimulation media (p < 0.01, p < 0.05 respectively). There was also an increasing number of ASC specks found within the lysates when the cells were treated with nigericin over time. Lysate specks were present at 7.5 min and markedly increased by 30 min (p < 0.05), but there were also two significant reductions in the number of lysate ASC specks at 15 min (~2-fold) and 60 min (~1.6-fold) nigericin (Fig. 1B). The ~2-fold decrease in the 15 min nigericin lysate specks matched to the 2-fold increase in the corresponding media (Fig. 1B). The assay was also validated with two specificity controls. The first one was when we applied the correct capture antibody for specific pulldown of ASC specks to the surface and detected the level of non-specific single-molecule fluorescence using a non-target Alexa647 IgG detection control antibody. As our second specificity control, we did not use a capture antibody and detected the non-specific binding with the correct fluorescent ASC antibody. We confirmed that both our specificity controls gave us significantly lower non-specific fluorescence compared to the sample (15 min nigericin-treated THP-1 lysates or media) with the correct capture and detection

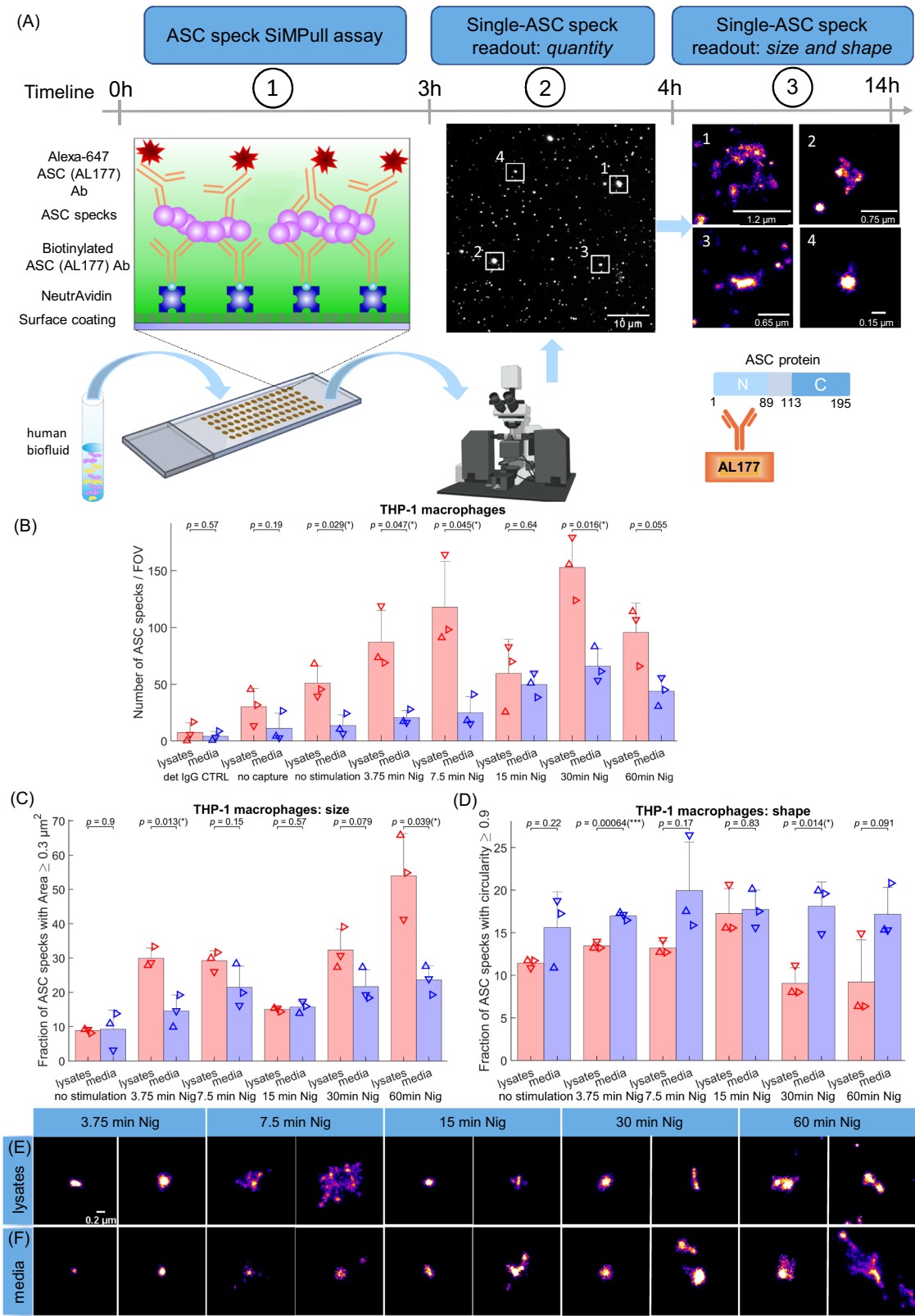

**Fig. 1 | Single ASC speck imaging single-molecule pulldown (SiMPull) assay.**
**A** Workflow of the SimPull platform for characterisation of the quantity, size and shape of ASC specks in human biofluids. **B** Time course quantification of ASC specks detected in the lysates and media from inflammasome-activated (LPS + Nigericin for 3.75, 7.5, 15, 30, 60 min) THP-1 macrophages vs non-stimulated controls (LPS only). Data points in B represent independent biological replicates (*n* = 3). Each replicate is an average of 12 different fields of view (images). The data are shown as mean ± SD. Permutation (exact) test: *\*p* < 0.05. **C, D** Comparison in

the fraction of individual ASC aggregates with (**C**) area ≥ 0.03 µm² and (**D**) circularity ≥ 0.9 detected in the conditioned lysates vs media with *d*STORM. Data points in (**C**) and (**D**) represent independent biological replicates (*n* = 3). Permutation (exact) test: *\*p* < 0.05, *\*\*\*p* < 0.001. **E, F** Example super-resolution (*d*STORM) images of ASC specks of different sizes and shapes detected in the THP-1 cell lysates and media (**C, D**). One out of three representative replicates is displayed here. Images were representative across experiments. Scale bar, 0.2 µm. Source data are provided as a Source Data file.

**Table 1 | Demographic and clinical characteristics of participants included in the PD versus control serum and CSF sample comparison**

| | Serum cohort 1 | | | Serum cohort 2 | | | CSF cohort | | |
|---|---|---|---|---|---|---|---|---|---|
| | Control | PD | *p*-value | Control | PD | *p*-value | Control | PD | *p*-value |
| Sample size | 10 | 10 | 1 | 5 | 8 | 0.6 | 6 | 6 | 1 |
| Age (years) | 64.8 ± 11.7 | 67.6 ± 7.8 | 0.5 | 57.7 ± 9.0 | 63.8 ± 6.9 | 0.2 | 66.1 ± 6.6 | 67.4 ± 7.0 | 0.8 |
| Sex (% male) | 10% | 60% | 0.1 | 40 % | 50% | 1 | 83% | 83% | 1 |
| ACE-III | - | 91.8 ± 6.6 | - | - | 93.8 ± 3.9 | - | 96.5 ± 2.1 | 93.2 ± 6.9 | 0.3 |
| Disease duration (years) | - | 0.5 ± 0.3 | - | - | 0.5 ± 0.3 | - | - | 1.3 ± 0.6 | - |
| Hoehn & Yahr | - | 1.6 ± 0.5 | - | - | 1.8 ± 0.7 | - | - | 2.0 ± 0.0 | - |
| MDS-UPDRS III | - | 26.8 ± 9.4 | - | - | 29.0 ± 15.2 | - | - | 31.4 ± 8.5 | - |
| MDS-UPDRS Total | - | 50.3 ± 17.5 | - | - | 50.3 ± 27.8 | - | - | 56.6 ± 14.8 | - |

Values represent the mean ± SD. Variables were compared using the permutation (exact) test except the sample size for which the binomial test was used (*p < 0.05).

antibody indicating that the measured signals are ASC specks and not non-specific (Fig. 1B). After successfully validating the assay, we applied this method for super-resolution imaging of ASC speck aggregates in the same THP-1 samples. We detected ASC aggregates ranging in size between 0.035 μm (area = 0.001 μm², see the data analysis and statistical testing section for details) to 1 μm (area = 0.745 μm²) in both cell lysates and secreted media, although the media had a trend for higher proportion of smaller and rounder ASC specks than the cell lysates (Supplementary Fig. 2 and Fig. 1C, D). At 7.5 min nigericin ASC formed specks of up to 1 μm in size (Fig. 1E) in the cell lysates which seem to increase further only after 60 min nigericin treatment (Fig. 1C). There was a significant difference in size distributions of the ASC specks between the 7.5 min and 15 min nigericin in both THP-1 cell lysates and media (Supplementary Fig. 2E, G). The shape of ASC specks was also significantly different between the 7.5 min and 15 min nigericin lysates but not for those media (Supplementary Fig. 2F, H). If one considers only ASC aggregates bigger than 0.2 μm (area > 0.03 μm²), cell lysates and those media from cells treated with 7.5 min nigericin had about 29% and 22% fractions of these aggregates compared to only 15% and 16% for the 15 min nigericin lysates and media (Fig. 1C). Both the 15 min nigericin cell lysates and media contained the smallest and most circular ASC specks (Fig. 1C, G).

Overall, these experiments showed that we could detect ASC speck formation in LPS-primed nigericin-activated THP-1 cells and that the ASC specks formed early in the inflammatory response (7.5 min nigericin) are as big as those treated with inflammasome activators for 4 times longer (30 min nigericin). We also observed a critical time point of 15 min with nigericin at which ASC formed the smallest and most circular specks.

## ASC specks in human biofluids as a biomarker of inflammation and inflammasome formation

Using samples from people with early PD (disease duration < 2 years), early AD, AD with moderate dementia and age/sex matched groups without neurological disease (HC), we then measured the number of ASC specks in CSF (*n* = 6 PD vs 6 HCs; *n* = 20 early AD vs 20 HCs) and serum (PD cohort 1: *n* = 10 PD vs 10 HC, PD cohort 2: *n* = 8 PD vs 5 HC; AD cohorts: *n* = 30 HC vs 20 early AD vs 20 AD with dementia). AD patients were assigned to 2 subgroups: early AD (*n* = 20, amnestic mild cognitive impairment or mild dementia severity, sampled at first visit to the memory clinic, positive for AD CSF biomarker profile) and AD with moderate dementia (*n* = 20, MMSE score = 18.4 ± 6.8, symptom duration = 4.5 ± 2.3 years). We also assessed soluble aggregates in samples derived from soaking brain tissue. Amygdala samples from PD cases (*n* = 9) with Braak Lewy Body stages from 3 – 6 were compared to amygdala samples from controls without known neurological disease during life (*n* = 5, Lewy body Braak stage 0, tau Braak stage 0, 1 or 2).

Demographic and clinical characteristics of participants and brain donors are summarized in Tables 1–3.

In both our early-stage PD and AD serum cohorts, we found a 1.5-4.2-fold increase in the number of ASC specks in people with disease compared to controls (Fig. 2A, B, D). To establish the utility of ASC specks as a potential fluid biomarker of inflammation, inflammasome formation and cell apoptosis, we performed a ROC curve analysis. We showed that the number of ASC specks in serum distinguished people with early PD from controls with AUC = 83% in cohort 1 and 100% in cohort 2 (Fig. 2C). Although the total number of detected ASC specks in CSF (Fig. 2G) was only half of those detected in serum, the number was increased 1.4-fold in PD CSF allowing us to fully discriminate PD from age-matched controls (no overlap between PD and control data) (Fig. 2G). In AD serum, we detected an increased quantity of ASC specks in the serum of AD patients at early disease stage compared to controls (Fig. 2D, *p* = 0.0044) although there was a large overlap in the data between these two groups giving an AUC of 64% (Fig. 2E). However, in serum samples from people in later stage AD with moderate stage dementia, we observed a significantly larger increase in the number of ASC specks by a factor of 4.2 in AD dementia serum compared to controls allowing us to differentiate controls from late-stage disease with an AUC of 88% (Fig. 2E). Similar to PD CSF, the number of ASC specks was increased 1.3-fold in the CSF of patients with early-stage AD compared to controls (Fig. 2F). The number of ASC specks was significantly positively correlated (*R* = 0.63, *p* = 0.0029) in paired serum and CSF samples from the same patients with early-stage AD (Supplementary Fig. 3F). Interestingly, CSF ASC speck number in people without a neurological condition was inversely correlated (*R* = −0.46, *p* = 0. 046) with number in serum (Supplementary Fig. 3F).

ASC specks were also detectable in human brain tissue. Since we are more interested in characterising extracellular ASC specks, we extracted the soluble aggregates from frozen brain tissue of the amygdala brain region from donors with PD and controls without neurological disease using a published protocol[32] (see the Methods section for details). In brief, soaked amygdala brain samples were prepared through incubating (soaking) pre-chopped 300 mg brain pieces in artificial CSF allowing protein aggregates diffuse (soak) into the solution and then collecting the supernatant after two sequential centrifugation steps. We found that the number of ASC specks in the soaked brain fluid samples was similar between controls and PD (Fig. 2H).

We also performed a series of control experiments to confirm that we were detecting ASC speck aggregates. As demonstrated in the Fig. 2J, K and H, the non-specific signal due to the sample sticking to the surface without capture antibody but with correct detection antibody was negligible in comparison to the sample signal in the presence of the correct capture and detection antibody as verified using PD serum

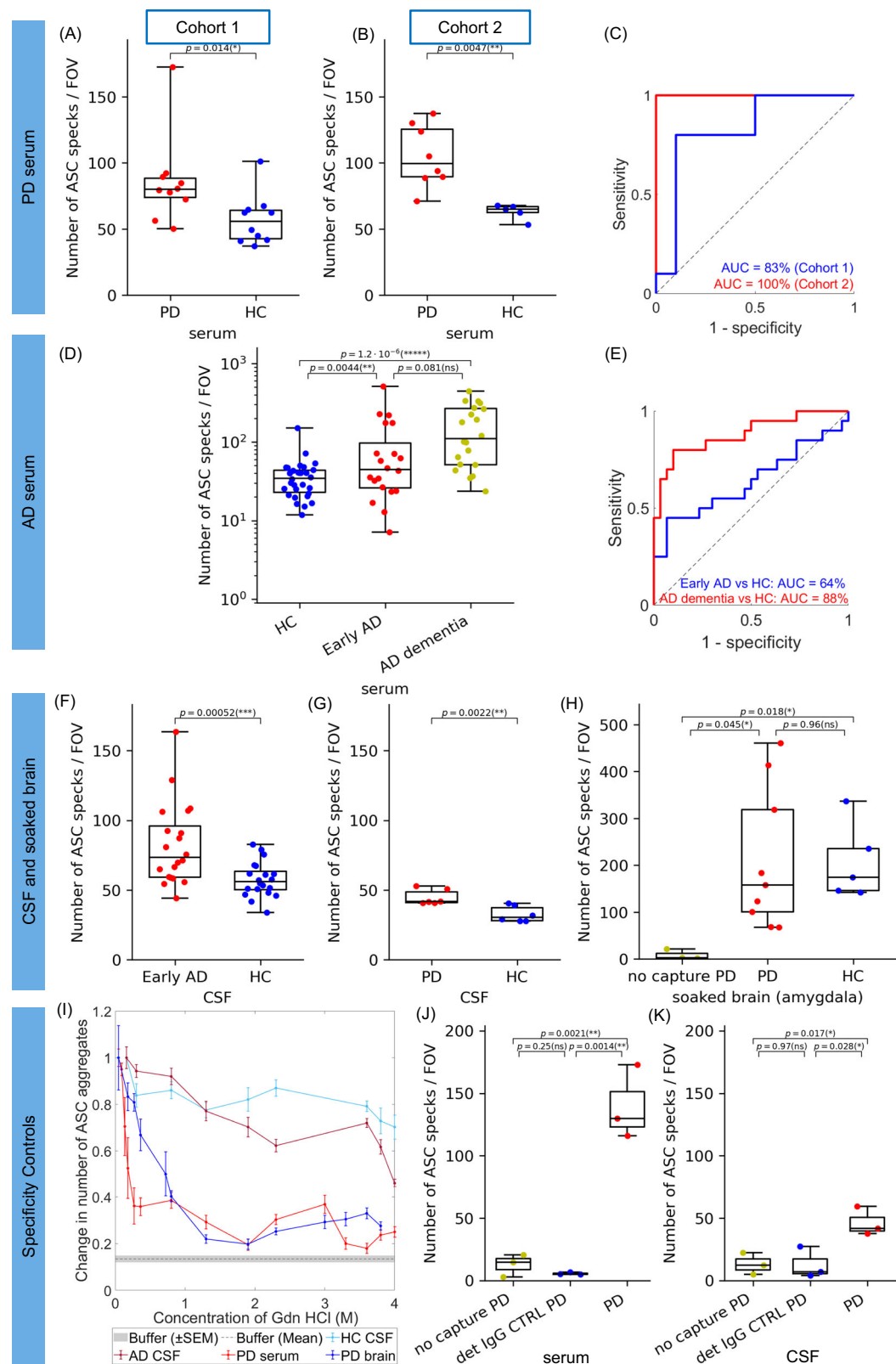

and brain samples. We also confirmed that our ASC-SiMPull assay has low non-specific fluorescence compared to a positive control sample (PD serum and CSF) indicating that most of the detected aggregates were ASC specks and not something else (Fig. 2J, K). As a cross-validation biochemical assay to prove our ASC SiMPull method is aggregate specific, we did a conformation stability assay on serum, CSF and brain homogenate samples. Prior to the SiMPull assay, samples

underwent a denaturation treatment with different concentrations of guanidine hydrochloride (Gdn HCl) which affects the secondary and tertiary structure of protein aggregates without affecting the primary structure of monomers. All samples were then diluted back to the same low concentration of Gdn HCl (see the Methods section for details) and analysed using SiMPull for ASC speck detection. Since only aggregates in the sample can be denatured by treatment, we observed

**Fig. 2 | Detection of ASC specks in human serum, CSF and soaked post-mortem brain samples from people with PD and AD compared to age-matched controls using SiMPull assay. A–H** Quantification of the number of ASC specks per field of view (FOV) detected in serum from (**A**, **B**) two cohorts of PD patients vs controls and (**D**) a cohort of patients with early AD, and AD dementia vs cognitively unimpaired controls. Quantification of the number of ASC specks per FOV detected (**F**) in CSF from early AD patients vs controls, (**G**) in CSF from PD patients vs controls and (**H**) in the amygdala of soaked post-mortem brains from 9 patients with PD versus 5 cognitively unimpaired controls. Data are presented as box plots (centre line at the median, upper bound at 75th percentile, lower bound at 25th percentile) with whiskers at minimum and maximum values. Each dot represents one participant. Permutation (exact) test: *$p < 0.05$, **$p < 0.01$, ***$p < 0.001$,****$p < 0.00001$.

**C**, **E** ROC curve analysis for quantity detection of ASC specks in human biofluids as a promising marker of inflammation discriminating controls from people with neurodegenerative disease. **I** Denaturation curve of ASC specks present in PD human brain homogenate ($n = 1$ PD), PD serum ($n = 1$ PD), AD CSF ($n = 1$ AD) and HC CSF ($n = 1$ HC) samples when treated with increasing concentration of Gdn HCl varying from 0.05 to 4 M. Error bars are mean ± SEM from $n = 12$ FOVs. One representative replicate out of two is displayed here. (**H**–**K**) Specificity controls for ASC speck detection in SiMPull assay using a no capture and correct detection antibody (no capture) and a correct capture and non-target IgG isotype control detection antibody (detection IgG CTRL) for PD serum (**J**), CSF (**K**) and soaked brain samples (**H**) from $n = 3$ donors each. Two-tailed *t*-test: *$p < 0.05$. Source data are provided as a Source Data file.

an exponential decay in the number of detected ASC aggregates with increasing concentrations of Gdn HCl (Fig. 2I). This confirmed that we are specifically detecting aggregates. To confirm our ASC SiMPull assay is chemically specific to the ASC protein detection, we measured the number of ASC aggregates before and after immunoprecipitation of ASC protein from two different PD serum samples (see the Methods section for details) and observed an almost complete depletion of ASC aggregates from the samples (Supplementary Fig. 3G). This confirms our ASC SimPull assay detects ASC protein aggregates present in serum.

For comparison with our ASC speck measurements, we also measured the concentration of a panel of cytokines (IL-1β, IFN-γ, IL-2, IL-4, IL-6, IL-10, IL-12p70, IL-17A, TNF-α) and C-reactive protein (CRP) levels in our cohort of 20 AD with moderate dementia and 10 HC serum samples (see the Methods section for details). We had insufficient samples available to perform similar assays in our cohort of early AD subjects. IL-1β is an established pro-inflammatory marker which is released by cells following inflammasome activation and can be measured in cell-free supernatant as a read-out of cellular inflammation in vitro[33]. In the serum samples, there was a strong positive correlation of our ASC speck marker with IL-1β levels in both AD and HC serum samples ($R = 0.73$, $p = 0.021$ (*) in HC serum, $R = 0.28$, $p = 0.23$ in AD serum, $R = 0.43$, $p = 0.019$ (*) when combined AD and HC serum samples together) (See the Supplementary Fig. 7A, B). ASC specks also strongly correlated with the levels of CRP in HC serum samples only ($R = 0.76$, $p = 0.017$ (*)), IL-2 ($R = 0.67$, $p = 0.033$ (*) in HC serum, $R = -0.26$, $p = 0.27$ in AD serum) and IL-10 in HCs only ($R = 0.79$, $p = 0.0065$ (**) in HC serum, $R = 0.3$, $p = 0.2$ in AD serum) (See the Supplementary Fig. 7C–E). There were no significant correlations of serum ASC speck levels with other measured cytokines. We also measured the levels of IL-1β, IFN-γ, IL-2, IL-4, IL-6, IL-10, IL-12p70, IL-17A, TNF-α cytokines in PD and age/sex matched HC serum samples where sufficient sample was available ($n = 7$ PD, $n = 11$ HC). We found that the serum levels of IL-1β positively correlated with ASC speck levels HCs ($R = 0.54$, $p = 0.088$) (See the Supplementary Fig. 8A, D) and in PD and HC samples combined ($R = 0.43$, $p = 0.075$) but not in the PD serum samples alone ($R = -0.3$, p = 0.52), likely due to the small sample size ($n = 7$ PD cases). ASC specks also correlated with the levels of IL-17A in HCs and PD patients ($R = -0.7$, $p = 0.021$ (*) in HC serum, $R = 0.78$, $p = 0.037$ (*) in PD serum), IL-10 in PD patients only ($R = 0.74$, $p = 0.058$ in PD serum), IL-4 ($R = 0.45$, $p = 0.06$ in HC serum) and IL-12p70 in HCs only ($R = -0.61$, $p = 0.05$ (*) in HC serum) (See the Supplementary Fig. 8B, E, G, H, I). There was no significant correlation of the ASC speck marker with the IFN-γ, IL-2, IL-6 and TNF-α cytokines in the analysed PD and HC serum samples.

Overall, the cytokine which was correlated most consistently with ASC speck levels was IL-1β, as might be anticipated given the known mechanistic relationship between inflammasome activation and IL-1β production. Serum ASC speck levels showed a better dynamic range (-2-4 fold increase in AD and PD serum compared to controls) than IL-1β [34,35] (1.3-1.4 fold increase in AD and PD serum), and greater sensitivity

to differentiate PD/AD from control serum (ASC speck: AUC = 89% AD vs HC serum, AUC = 87% PD vs HC serum compared to IL-1β: AUC = 66% AD from HC serum, AUC = 73% PD from HC serum).

## ASC speck composite serum biomarkers for AD and PD

ASC specks are not specific for AD and PD since they are a general marker of inflammasome formation. However, given that AD and PD are complex multifactoral disorders with different protein aggregates contributing to disease, ASC specks may have a better diagnostic value when combined with other protein aggregate markers of AD/PD. To test the potential of our ASC speck blood assay for improving discrimination between AD samples and controls, we combined measurement of ASC specks with measurements of total Aβ aggregates, and phosphorylated tau aggregates in the same set of serum samples from 20 early AD, 20 AD with moderate dementia patients and 30 controls using our established SiMPull assay with a pair of identical 6E10 antibodies for capture and detection of Aβ aggregates and the AT8 antibody pair for phosphorylated tau (p-tau) at positions 202/205 (p-tau-AT8) (Fig. 3A–D). We demonstrated that the ratio of the number of ASC specks (*ASC*) to Aβ aggregates (*Aβ*) detected in serum, *ASC/Aβ*, (Fig. 3A) was 10.4 and 12.5 times larger in early AD and late AD (AD dementia) than the control serum and the discrimination accuracy of the assay compared with measurement of Aβ alone improved from 66% (AUC = 66% using Aβ aggregates only, Supplementary Fig. 3D) to 78% in early AD (AUC = 78%, Fig. 3C) and from 59% (AUC = 59% using Aβ aggregates only, Supplementary Fig. 3D) to 89% in AD with moderate dementia (AUC = 89%, Fig. 3C). Also, the use of the number p-tau-AT8 aggregates as a single biomarker in serum gave us a low diagnostic accuracy of 55% (AUC = 55%) in early AD and 74% in AD with moderate dementia (Supplementary Fig. 3A, C). By combining the measures of serum Aβ aggregates, p-tau-AT8 aggregates and ASC specks together we built a composite biomarker profile defined as a ratio of the sum of the number of p-tau-AT8 aggregates and ASC specks to the number Aβ aggregates, *(p-tau-AT8 + ASC) / Aβ* (Fig. 3B), which was increased 6.2-fold in early AD and 11.7-fold at later stage (AD with dementia) and could distinguish early AD from control serum with an accuracy of 92% (AUC = 92%) and AD moderate dementia from control with an accuracy of 95% (AUC = 95%) (Fig. 3E). Removal of ASC speck measurements from the *(p-tau-AT8 + ASC) / Aβ* ratio gave us a significantly reduced AUC of 81% in early AD serum and a slightly lower AUC of 93% in AD moderate dementia (Supplementary Fig. 3E).

We also explored different combinations of the number of ASC specks, α-syn aggregates (*α-syn*) and Aβ aggregates (see the data analysis and statistical testing section for details) in serum from the two early PD cohorts (Fig. 3) which were assayed independently (see Table 1 and the participants section for details). In both cohorts we found that the serum *(α-syn + ASC)/Aβ* ratio (Fig. 3J, K) was increased 3.1 – 4.9-fold in disease (cohort 1 and 2) and accurately differentiated people with early PD from controls (AUC = 97% in cohort 1 and 93% in cohort 2, Fig. 3L). Interestingly, the *ASC/Aβ* ratio (Fig. 3G, H) alone also gave an AUC of 96% and 3.4-fold increase in cohort 1 and AUC of

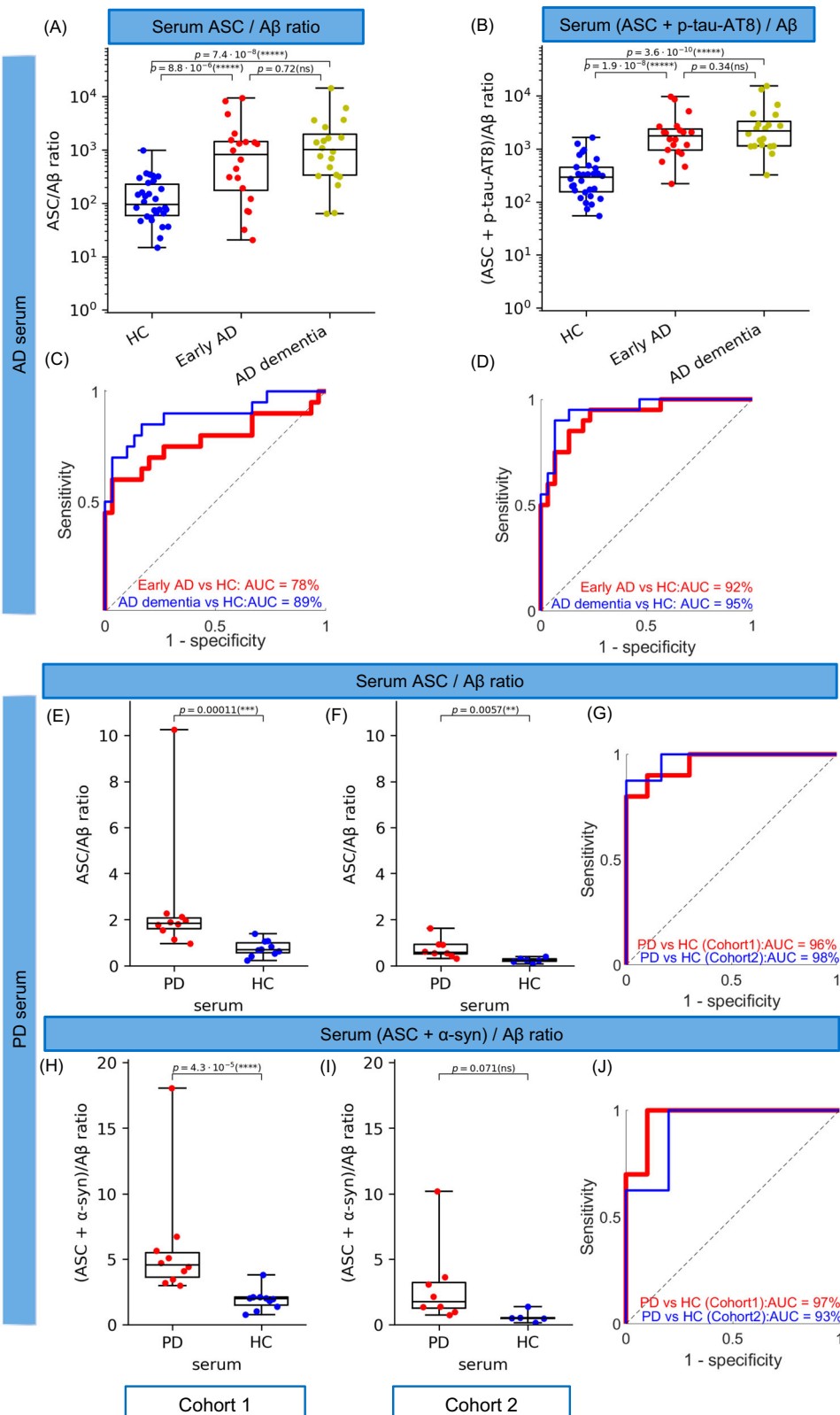

**Fig. 3 | Combination of single-aggregate measurements of ASC specks with Aβ aggregates and p-tau-AT8 aggregates phosphorylated at positions 202/205 in early AD (n = 20), AD with dementia (n = 20) and HC (n = 30) serum samples and with α-syn aggregates and Aβ aggregates in the two early-stage PD serum cohorts (cohort 1: n = 10 PD and 10 HC and cohort 2: n = 8 PD and 5 HC). A** *ASC/Aβ* and (**B**) *(ASC + p-tau-AT8) /Aβ* ratios as candidate composite biomarkers in AD serum; (**E**, **F**) *ASC/Aβ* and (**H**, **I**) *(ASC + α-syn) /Aβ* ratios as candidate composite

biomarkers in early-stage PD serum. The data in (**A**, **B**) are plotted in log10 scale. Data are presented as box plots (centre line at the median, upper bound at 75th percentile, lower bound at 25th percentile) with whiskers at minimum and maximum values. Each dot represents one participant. Permutation (exact) test: *p < 0.05, ***p < 0.001,****p < 0.0001,*****p < 0.00001. **C–J** ROC curve analysis using the corresponding metrics. Source data are provided as a Source Data file.

98% and 3-fold increase in cohort 2 (Fig. 3I). The individual single-molecule markers of α-syn and Aβ used for calculating ratios for PD and HC serum groups in both cohorts are shown in Supplementary Fig. 4H–K.

Overall, these results show that by combining ASC specks with measurements of other aggregates in the serum to form a composite biomarker achieves excellent discrimination and a larger dynamic range between patients and controls. For PD, *(α-syn + ASC)/Aβ* seems most promising (in cohort 1: AUC = 97% and 3.1-fold increase, in cohort 2: AUC = 93% and 4.9-fold increase) and for early AD the serum *(p-tau-AT8 + ASC)/Aβ* ratio gives excellent discrimination (AUC = 92%) and a dynamic range of 6.2.

### Size and shape of ASC specks in biofluids of people with PD/AD as a biomarker

We super-resolved ASC specks in human PD and AD biofluids and controls to extract information about the size and shape of individual aggregates. We used this information as additional metrics which may allow us to detect larger differences between controls and people with early disease and increase the accuracy of aggregate-based biomarkers even further. In our previous work, we showed that by combining the ratio-metric measurements of α-syn to Aβ with the size and shape of α-syn aggregates imaged with *d*STORM we could improve the accuracy for distinguishing PD from control serum from 85% – 90% AUC[29]. Individual ASC aggregates are characterised by their area, perimeter, and circularity. Cumulative histograms of the area, perimeter and circularity distributions and their relative differences can be generated to investigate the morphological differences of the aggregates. By combining single-aggregate ASC speck imaging (SiMPull) with *d*STORM, we can distinguish between aggregates of different sizes bigger than 30 nm (our resolution limit) which are the majority of our detected aggregates in serum and brain samples (~99%). We have performed control experiments to confirm that 8k frames in *d*STORM are sufficient to accurately measure the size and shape of ASC specks in human biofluids. As demonstrated in the Supplementary Fig. 6, there is no significant change (with a 95% confidence) in the size and shape distributions of ASC aggregates in serum when imaged using 16 k frames instead of 8k frames. When compared with the interquartile ranges (IQR) [25%–75%] of the size and shape distributions imaged with 2 k, 5 k, 8 k, 11 k and 14 k frames, the morphological information on ASC aggregates started to have a stable distribution with 8 k frames (area IQR = 0.015 μm² and circularity IQR = 0.287, Supplementary Fig. 9). There was also no significant photobleaching of all fluorophores after 8k frames, as evidenced by sufficient number of localisations we detected from 8 k-16 k frames (Supplementary Fig. 6E). This indicates that the morphology measurements of ASC aggregates saturate at 8k frames due to sufficient sampling (limited by the number of fluorophores on each aggregate).

To characterise ASC specks accumulated in the brain, we super-resolved soluble ASC specks extracted from post-mortem human brain of PD patients. We found that ASC specks ranged in size between 0.035 μm (area = 0.001 μm²) to 0.37 μm (area = 0.108 μm²) (Fig. 4E) and they were smaller and rounder in PD than non-demented control brain (Fig. 4A). To check the trend was consistent across all patients, we examined individual aggregate trends for each patient and observed a clear separation in size (Supplementary Fig. 4G–J) and shape (Supplementary Fig. 4K, L) distributions of ASC specks between each patient and the controls. There were 12% and 8% higher proportions of ASC specks with area < 0.018 μm² (*p* = 0.0028, Supplementary Fig. 4A) and with circularity larger than 0.8 (*p* = 7.4·10⁻⁵, Supplementary Fig. 4C) detected in PD compared to control brains representing the maximum fractional difference. By combining both parameters together (the area ≤ 0.018 μm² and circularity ≥ 0.8) we observed 8% higher proportion of ASC specks < 0.15 μm and rounder in PD soaked brain compared to control soaked

brain samples (*p* = 0.0015, Fig. 4A). To relate to the ASC specks present in the periphery, we characterised the size and shape of ASC specks in serum (Figs. 5, 6). We found that ASC specks in both early PD (Fig. 5A) and AD (Fig. 6A) serum ranged in size between 0.035 μm (area = 0.001 μm²) to 0.36 μm (area = 0.1 μm²) (Figs. 5, 6G) with a higher proportion of smaller and rounder ASC specks than controls, similar to those in the brain (Fig. 4A). By combining the proportion of ASC specks that were smaller and rounder than a defined area and circularity threshold (area ≤ 0.05 μm² and circularity ≥0.5 in PD serum cohort and area <0.04 μm² and circularity > 0.7 in AD serum cohort) we observed 2% and 7% higher proportion of these ASC specks in PD (*p* = 0.0019, Fig. 5A) and AD (*p* = 0.003, Fig. 6A) than in control serum. When converting area into the size, Alzheimer's and Parkinson's patients have a significantly higher fraction of ASC specks smaller than 0.226 and 0.252 μm, respectively, in serum, than people without disease.

We then made a combined threshold based on the proportion of aggregates with an area smaller and circularity higher than the thresholds (morphologically distinctive ASC specks) established above. This allowed us to distinguish controls from people with disease with an excellent accuracy (AUC = 100% in PD human brain (Fig. 4B), AUC = 94% in early PD serum (Fig. 5B) and AUC = 90% in AD serum (Fig. 6B)). We have also been able to distinguish small but significant differences in size and shape of ASC specks between PD and AD serum (Supplementary Fig. 5). There was about 2.5% higher proportion of smaller and rounder ASC aggregates with area ≤ 0.0021 μm² and circularity ≥ 0.5 (Supplementary Fig. 5A) or perimeter ≤ 0.145 μm and circularity ≥ 0.5 (Supplementary Fig. 5D) present in serum of PD patients compared to AD. This discriminative area (0.0021 μm²) or perimeter (0.145 μm) is equivalent to the size of ASC aggregates of ~ 0.05 μm with more of them present in PD than AD serum. When setting these parameters of area and circularity as our combined threshold we can distinguish PD from AD diagnosis with an accuracy of 82% (AUC = 82%, Supplementary Fig. 5B). We have been also able to improve the discrimination accuracy to 85% (AUC = 85%, Supplementary Fig. 5E) using a combination of the defined perimeter and circularity thresholds for ASC specks (perimeter ≤ 0.145 μm and circularity ≥ 0.5).

Finally, we explored whether a combination of morphology of ASC specks together with the quantity of ASC specks and disease-specific protein aggregates measured in our SiMPull assay could further improve diagnostic biomarker performance (Figs. 5, 6D). We took the proportion of morphologically distinctive serum ASC specks established above and added them to a panel of the number of ASC specks, α-syn aggregates and Aβ aggregates in PD serum and ASC specks, p-tau-AT8 aggregates and Aβ aggregates in AD serum to explore their different combinations using an unbiased approach (see the data analysis and statistical testing section for details). We found that the total number of ASC specks divided by the morphologically distinct fraction of ASC specks (total ASC/morphologically distinct fraction ASC ratio) increased 4.7-fold in disease and worked best at differentiating AD from controls (AUC = 100%, Fig. 6E). In PD (cohort 1), the ratio of the sum of the morphologically distinct fraction of ASC specks and total number of ASC speck to the number of Aβ aggregates *((morphologically distinct fraction ASC + total ASC)/Aβ)* was the most promising at discriminating early PD from control serum giving us AUC of 97% (Fig. 5E) and increased 1.7-fold in PD (super-resolution imaging of ASC specks was not performed for PD cohort 2).

### Higher-throughput morphology analysis of ASC specks in human biofluids using faster *d*STORM

In the previous section, we performed *d*STORM (*n* = 8000 frames) in non-overlap camera mode with 30 ms exposure time per frame and subsequent readout time of ~ 33.7 ms which doubled the imaging time

**Fig. 4 | Morphology analysis of ASC specks extracted from the amygdala of human PD ($n$ = 9) and non-demented control ($n$ = 5) post-mortem soaked brain using $d$STORM. A** Comparison of the fraction of individual ASC aggregates that are smaller (area $\leq$ 0.018 µm²) and rounder (circularity $\geq$ 0.8) than the defined threshold in the brain of people with PD vs controls. We identified the area (size) and circularity (shape) threshold (maximum statistically significant difference in the size and shape of ASC speck histograms between diagnostic groups) giving us the morphological phenotype of ASC specks which is increased in PD brains. Data are presented as box plots (centre line at the median, upper bound at 75th percentile, lower bound at 25th percentile) with whiskers at minimum and maximum values.

Each dot represents one participant. Permutation (exact) test: **$p$ < 0.01. **B** ROC curve analysis of the identified phenotype. **C**, **D** Examples of super-resolved ASC aggregates in PD (**C**) and control (**D**) brain samples. One out of three representative replicates is displayed here. Images were representative across experiments. Cumulative size (**E**) and shape (**G**) distributions of ASC specks for PD vs control brains. Difference between PD and control cumulative size (**F**) and shape (**H**) distributions retrieved from (**E** and **G**). The dotted line indicates 99% confidence using the Kolmogorov-Smirnov statistical test. Source data are provided as a Source Data file.

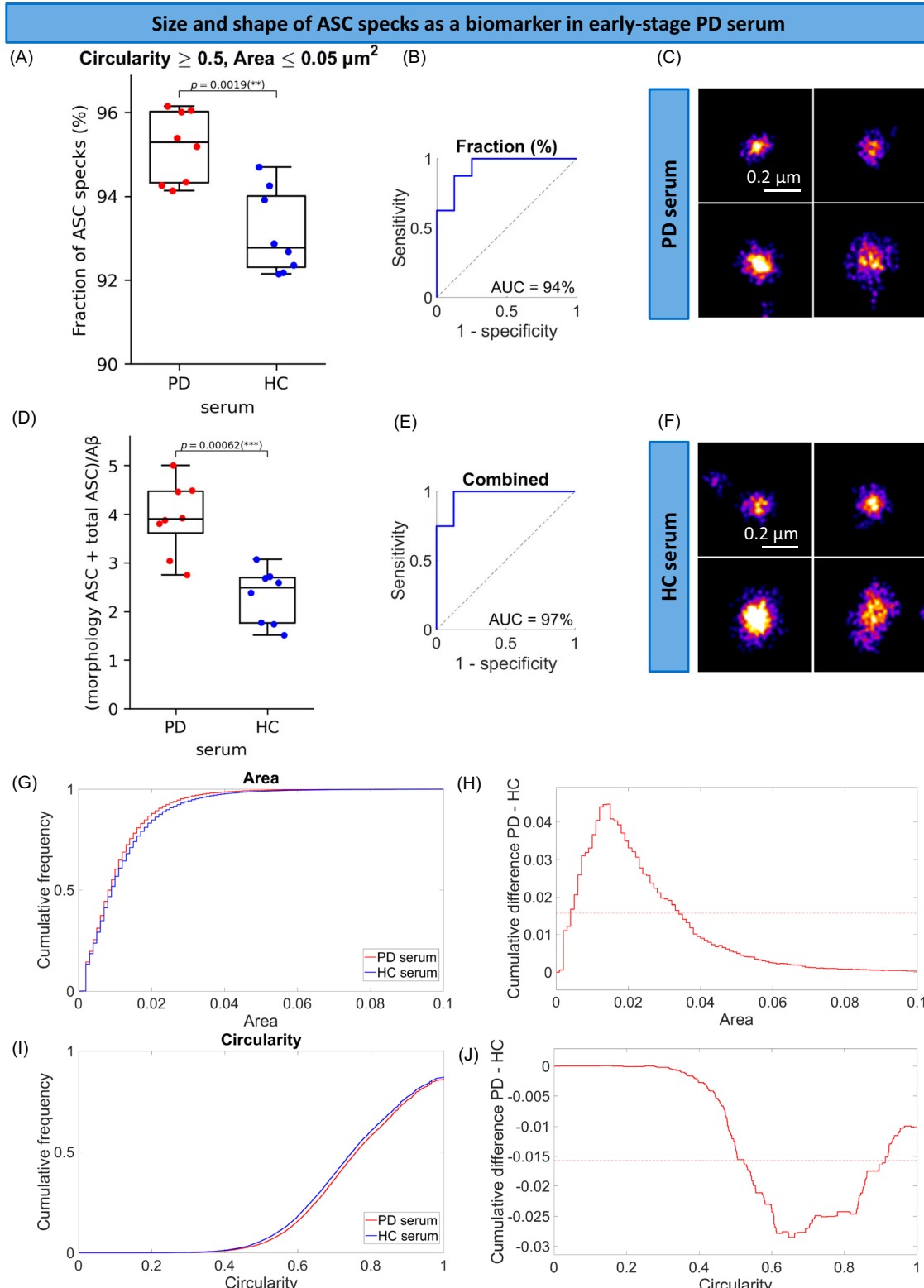

**Fig. 5 | Morphology analysis of ASC specks detected in the control (*n* = 8) and PD (*n* = 8) serum from people with early-stage disease (within 6 months of diagnosis) using *d*STORM. A** Comparison in the fraction of individual ASC aggregates that are smaller (area ≤ 0.05 μm²) and rounder (circularity ≥ 0.5) than the defined threshold in the serum of people with PD vs controls. **B** ROC curve analysis of the identified phenotype. **D**, **E** The *(morphologically distinctive ASC fraction + total ASC)/Aβ* as a candidate composite biomarker in PD serum and its ROC curve analysis. Data are presented as box plots (centre line at the median, upper bound at 75th percentile, lower bound at 25th percentile) with whiskers at minimum and maximum values. Each dot represents one participant. Permutation (exact) test: ****p* < 0.001. **C**, **F** Examples of super-resolved ASC aggregates in PD (**C**) and control (**F**) serum samples. The examples selected here are representatives of all three replicates. Cumulative size (**G**) and shape (**I**) distributions of ASC specks for PD vs control serum. Difference between PD and control cumulative size (**H**) and shape (**J**) distributions retrieved from (**G**) and (**I**). The dotted line indicates 99% confidence using the Kolmogorov-Smirnov statistical test. Source data are provided as a Source Data file.

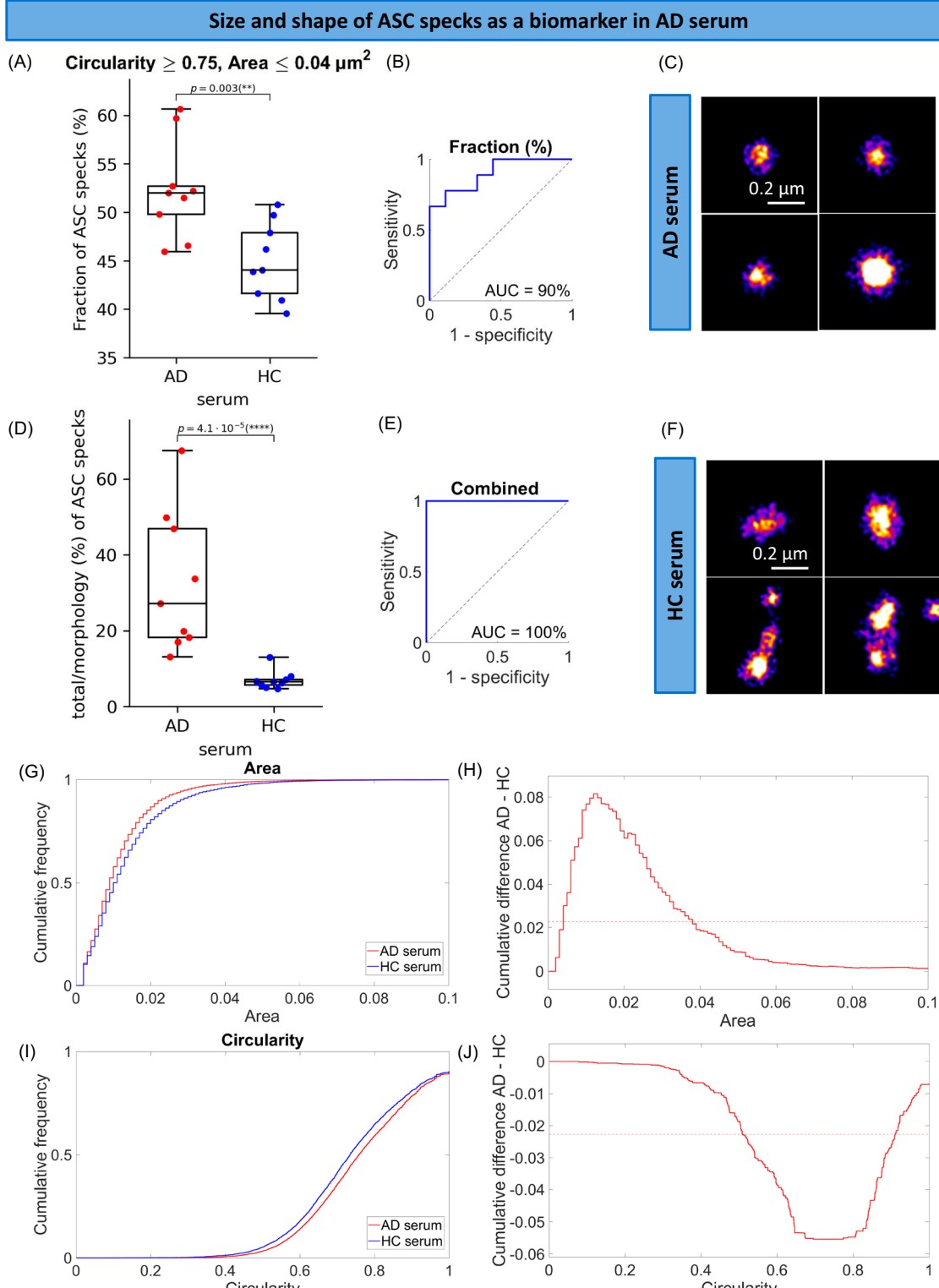

**Fig. 6 | Morphology analysis of ASC specks detected in the control ($n$ = 9) and AD dementia ($n$ = 9) serum using $d$STORM. A** Comparison in the fraction of individual ASC aggregates that are smaller (area ≤ 0.04 μm²) and rounder (circularity ≥ 0.75) than the defined threshold in the serum of people with AD vs controls. **B** ROC curve analysis of the identified phenotype. **D, E** The total ASC divided by the morphologically distinct fraction of ASC specks as a candidate composite biomarker in AD serum and its ROC curve analysis. Data are presented as box plots (centre line at the median, upper bound at 75th percentile, lower bound at 25th percentile) with whiskers at minimum and maximum values. Each dot represents one participant. Permutation (exact) test: **$p$ < 0.01, ****$p$ < 0.0001. **C, F** Examples of super-resolved ASC aggregates in AD (**C**) and control (**F**) serum samples. The examples selected here are representatives of all three replicates. Cumulative size (**G**) and shape (**I**) distributions of ASC specks for AD vs control serum. Difference between AD and control cumulative size (**H**) and shape (**J**) distributions retrieved from (**G**–**I**). The dotted line indicates 99% confidence using the Kolmogorov-Smirnov statistical test. Source data are provided as a Source Data file.

and therefore may be a potential disadvantage if the method would be used for diagnostics. To make the method more scalable, we performed faster *d*STORM imaging measured using the overlap camera mode (simultaneous exposure-readout) and compared its performance with the non-overlap mode. For both modes, we used 33 ms of

exposure time and the same number of frames of 8000. We found that both camera modes gave us similar results for the size and shape distributions of ASC specks in the CSF samples (within 95% confidence interval) and similar average lateral uncertainty of $13 \pm 5$ nm (Supplementary Fig. 11).

Since there was no difference in the ASC speck quantification but the data were acquired at double the imaging speed, we then explored the overlap mode for *d*STORM analysis of ASC specks in 14 early-stage AD and 14 age/sex-matched control human CSF from our study cohort (Table 2). Applying the same analysis approach as in the previous section, we found a significant increase in the number and proportion of smaller and rounder ASC specks (with area $\leq 0.03 \, \mu m^2$ and circularity $\geq 0.5$) in early AD compared to control CSF (Fig. 7A, B). Both of the defined metrics allowed us to distinguish early AD from controls CSF with an excellent accuracy giving AUC of 98% when used the metric of (the number of ASC specks with area $\leq 0.03 \, \mu m^2$ and circularity $\geq 0.5$) (Fig. 7D) and AUC of 93% for the metric of (the fraction of ASC specks with area $\leq 0.03 \, \mu m^2$ and circularity $\geq 0.5$) (Fig. 7E). The use of the overlap mode in our *d*STORM improved the imaging speed ~2 times and demonstrated that it was possible to measure biologically significant changes in the morphology of ASC specks at early disease stages. This is proof-of-concept that this method has potential for early disease diagnosis.

**Table 2 | Demographic and clinical characteristics of participants included in the early AD versus control serum and the AD dementia versus control serum sample comparison**

| Serum cohort | | | |
|---|---|---|---|
| | Control | Early AD | AD dementia |
| Sample size | 30 | 20 ($p = 0.2$) | 20 ($p = 0.2$) |
| Age (years) | $69.8 \pm 10.2$ | $73.3 \pm 6.7$ ($p = 0.2$) | $65.8 \pm 8.8$ ($p = 0.1$) |
| Sex (% male) | 53% | 60% ($p = 0.8$) | 55% ($p = 1$) |
| ACE-R | - | - | $56.4 \pm 20.2$ |
| MMSE | - | - | $18.4 \pm 6.8$ |
| Disease duration (years) | - | 0 (baseline) | $4.5 \pm 2.3$ |
| Matched CSF (n) | 20 | 20 | 0 |

Values represent the mean ± SD. *P*-value for the age and sex sample comparison of early AD /AD dementia group versus controls are shown in brackets. Variables were compared using the permutation (exact) test (*$p < 0.05$).

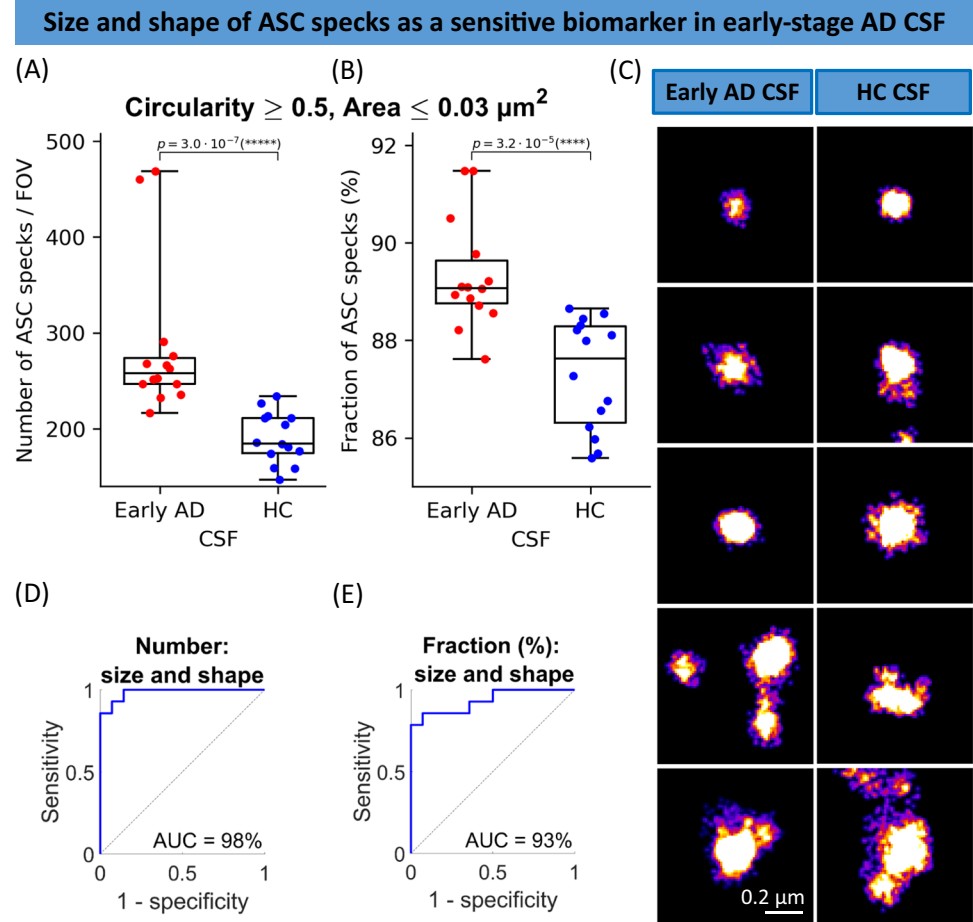

**Size and shape of ASC specks as a sensitive biomarker in early-stage AD CSF**

**Fig. 7 | Morphology analysis of ASC specks detected in the control ($n = 14$) and early-stage AD ($n = 14$) CSF using *d*STORM. A, B** Comparison in the number (**A**) and fraction (**B**) of individual ASC aggregates that are smaller (area $\leq 0.03 \, \mu m^2$) and rounder (circularity $\geq 0.5$) than the defined threshold in the CSF of people with early AD vs age-matched controls. **D, E** ROC curve analysis of the identified phenotype. Data are presented as box plots (centre line at the median, upper bound at 75th percentile, lower bound at 25th percentile) with whiskers at minimum and maximum values. Each dot represents one participant. Permutation (exact) test: ***$p < 0.001$, ****$p < 0.0001$. **C** Examples of super-resolved ASC aggregates in AD and control CSF samples. The examples selected here are representatives of all three replicates. Source data are provided as a Source Data file.

## Discussion

Immune system dysregulation and neuroinflammation are major pathophysiological drivers of disease progression in AD[36] and PD[25]. We introduce a new assay that has potential as an ultrasensitive biomarker of inflammation in the blood of people with AD and PD. The single-molecule pull-down detection assay was combined with *d*STORM, enabling measurement of the size (to 30 nm resolution) and shape of individual extracellular inflammasome ASC specks in the serum, as well as in the CSF and brain samples from people with PD and AD.

We developed a high-sensitivity antibody-based assay for detection of individual ASC specks in human biofluids. The quantity of extracellular, secreted ASC specks measured in the blood and CSF has utility as a biomarker of inflammation and inflammasome formation for PD and AD. By combining the quantity of ASC specks with measures of Aβ, α-syn and p-tau aggregates into different ratios in serum, we achieved high diagnostic accuracies of 92% for early AD and 95% for late AD ((p-tau-AT8 + ASC)/Aβ ratio) and of up to 98% for PD (both α-syn + ASC)/Aβ and ASC/Aβ. The use of ratios normalised to Aβ levels instead of absolute aggregate quantities helped us to achieve greater biomarker performance and a larger dynamic range between patients and controls (from 1.5 to 1.7 fold increased single ASC speck biomarker to 3-5 fold increased (α-syn + ASC)/Aβ ratio in PD serum compared to controls and from 1.5-fold increased single ASC speck biomarker to 6.2-fold increased (p-tau-AT8 + ASC)/Aβ ratio in early AD serum compared to controls). These combined biomarkers suggest that there is increased inflammation and production of α-syn aggregates in PD and increased inflammation and production of tau aggregates in AD. There is little change in the number of Aβ aggregates in serum in either disease allowing the number of Aβ aggregates to be used for normalisation. The increased accuracy of these new ASC speck composite biomarkers in discriminating PD and AD cases which were close to diagnosis from age-matched controls, together with validation in an independent group of samples, suggests that these markers may show promise as an early diagnostic biomarker for PD. Future studies using blood samples from patients with prodromal disease will be important to assess whether the marker can accurately predict conversion to manifest disease, whilst longitudinal studies are needed to assess whether inflammation changes during disease progression and in response to future disease modifying drugs can be detected.

By super-resolving ASC specks, we detected aggregates ranging in size from 0.035 μm to about 0.36 μm in both serum and brain. The soluble ASC aggregates from the post-mortem brain were present in similar morphology to those detected in serum, but PD patients had a significantly higher proportion of smaller and rounder ASC aggregates than controls. In comparison, in the THP-1 cell experiment with nigericin stimulation for 3.75, 7.5, 15, 30 and 60 min there was an ~30% reduction in the number of ASC specks detected in the cell lysates at 15 min nigericin accompanied by high levels of released ASC specks in the matched media. This suggest that pyroptosis of significant proportion of cells occurs by this early time point. Morphologically ASC specks at 15 min post nigericin were the smallest and roundest compared to other time points. This correlates with the morphology of ASC specks detected in disease samples. This may suggest that the increased number of smaller, rounder ASC specks we see in the disease state are from cells dying due to inflammasome activation in disease. For diagnostic potential, measuring the proportion of ASC specks that are smaller and rounder than threshold (morphologically distinctive ASC specks) can distinguish people with early disease from controls with an excellent accuracy (AUC = 94% in PD serum and AUC = 92% in AD serum). ASC speck morphological information has an even better diagnostic value when combined with the quantity of ASC specks: the total number of ASC specks divided by the morphologically distinct fraction of ASC specks (total ASC/morphologically distinct fraction ASC) in AD serum had an accuracy (AUC) of 100% and the (morphologically distinct fraction ASC + total ASC)/Aβ in PD serum had an

accuracy (AUC) of 97%. It will be important to perform similar measurements on a larger number of samples in future work to confirm these findings.

Collectively, our data support the hypothesis that the NLRP3 inflammasome is activated in early PD and AD, and lend further support to the idea that specific targeting of inflammasomes and/or their downstream constituents such as ASC, caspase 1 or cleaved gasdermin D[37] warrants testing as a disease-modifying strategy for PD/AD[38].

Overall, our study presents a sensitive and specific method for effective measurements of ASC specks in human biofluids, with the main focus on blood as a convenient sample for diagnostics. Although we have shown that blood-based composite ASC speck and protein aggregate biomarkers are sensitive to early Alzheimer's and Parkinson's disease stages, future studies are needed to establish the sensitivity of the assays in prodromal stages such as mild cognitive impairment, a prodrome of AD, and Rapid Eye Movement Sleep Behavior Disorder (RBD), a prodrome of α-synucleinopathies[39]. Our assay can characterise 40 different samples/wells per plate with all measurements of quantity, size and shape of ASC aggregates taking 14 h of which 10 h are done remotely with automated image acquisition. However, assay automation will help to make this method widely available and feasible for use in the clinical setting. We are currently working on our assay automation for which we programme a pipette robot (Analytik Jena CyBio FeliX robot) to automatically prepare the surface coating, perform washing steps, add the surface capture antibodies, add the samples and then add the imaging antibodies. The imaging instrument costs are within the same price range as a gene sequencing machine (e.g.: Illumina MiSeq) and hence technology could be straightforwardly commercialised in collaboration with diagnostic companies and used in central labs.

## Methods

### Participants

Patients with idiopathic Parkinson's disease (within 1–2 years of diagnosis according to UK PD Brain Bank Criteria) and controls without neurological disease were enroled from the Parkinson's Disease Research Clinic at the John Van Geest Centre for Brain Repair, University of Cambridge/Cambridge University Hospitals NHS Trust, UK. Participants with Parkinson's disease were assessed using the Movement Disorder Society Unified Parkinson's Disease Rating Scale (MDS-UPDRS), and the Addenbrooke's Cognitive Examination (ACE-III or ACE-R). Parkinson's disease stage was determined using the Hoehn and Yahr scale (Table 1). PD Dementia was an exclusion criterion. Patients with Alzheimer's disease (AD, including the prodromal state of mild cognitive impairment) and participants without neurological disease were recruited in the GOLDeN Study (Genetics of Leucopathology, Dementia and Neurodegeneration at the Cambridge University Hospitals NHS Trust) (Table 2). Participants with mild cognitive impairment were followed clinically to confirm progression and/or have biomarker evidence of underlying Alzheimer's disease pathology. The participants also underwent cognitive testing such as the ACE-R and Mini−mental state examination (MMSE), with the MMSE score < 24 (30) taken as an indicator of dementia[40]. Patient samples from the GOLDeN Study analysed in this study were collected in the established dementia phase (n = 20 cases, MMSE score = 18.4 ± 6.8). Ethical approval was obtained from the East of England - Essex Research Ethics Committee (16/EE/0445), and East of England−Cambridge Central Research Ethics Committee (03/303, and 15/EE/0270). Patients with early-stage Alzheimer's disease included in this study (n = 20 cases) were recruited from memory clinics in Sweden where they sought medical advice for the first time and were diagnosed with AD based on a positive AD CSF biomarker profile, defined as a CSF Aβ42/p-tau181 ratio < 10.25 (as measured by Lumipulse G600II, Fujirebio). Paired serum and CSF samples from these early-stage patients were used, collected at their baseline assessment. The study protocol was

**Table 3 | Demographic and clinical characteristics of the PD and control brain donors**

| Diagnosis | Sex | Age | Lewy Body Braak Stage | Tau tangle Braak Stage | Disease duration (years) | Post-mortem interval (hours) | Cause of death |
|---|---|---|---|---|---|---|---|
| PD | Male | 71 | 6 | 2 | Not available | 64 | - |
| PD | Male | 85 | 6 | 2 | 18 | 8 | PD dementia |
| PD | Male | 73 | 5 | 2 | Not available | 57 | - |
| PD | Female | 78 | 5 | 2 | 5 | 68 | PD |
| PD | Male | 76 | 5 | 1 | 10 | 60 | Aspiration Pneumonia |
| PD | Male | 77 | 5 | 3 | 9 | 101 | Pneumonia |
| PD | Male | 88 | 6 | 3 | 23 | 44 | Pneumonia |
| PD | Male | 90 | 5 | 3 | Not available | 52 | - |
| PD | Male | 84 | 4 | 2 | 6 | 21 | Pneumonia |
| Control | Male | 72 | 0 | 2 | - | 39 | - |
| Control | Female | 65 | 0 | 0 | - | 5 | - |
| Control | Male | 39 | 0 | 0 | - | 69 | |
| Control | Male | 73 | 0 | 0 | - | 51 | - |
| Control | Male | 62 | 0 | 1 | - | 63 | - |

approved by the regional ethics committee at the University of Gothenburg. Informed consent was provided by all participants. Biological sex of participants was determined by self-report. Controls were selected to be age and sex-matched at group level to participants with disease.

**Exclusion criteria**

For patient samples, exclusion criteria included recent infections < 3 weeks, a history of chronic inflammatory or autoimmune conditions (e.g. Crohn's disease, systemic lupus erythematosus, rheumatoid arthritis), use of immunomodulatory/anti-inflammatory medications, and vaccinations within the last 3 weeks.

**Serum and CSF sampling**

Blood from participants was collected by venepuncture using 7.5 ml S-Monovette tubes, samples were left to clot at room temperature for 15 min before centrifuging at 2000 rpm for 15 min at room temperature. The supernatant (serum) was collected and stored at −80 °C until use. In order to minimise the impact of repeat frozen-thawed cycles, upon collection of samples, single-use 11 µL aliquots were prepared into protein low bind tubes (Protein LoBind® Tubes 0.5 mL, Eppendorf). Serum samples from two independent cohorts of 10 PD patients and 10 age-matched controls (PD cohort 1, Table 1), and 8 PD and 5 controls (PD cohort 2, Table 1) were assayed. Serum samples from 20 early AD, 20 AD with dementia patients and 30 cognitively unimpaired controls were also used in the study (Table 2). 20 early AD and 20 cognitively unimpaired controls had matched CSF samples.

Lumbar puncture was performed using standard sterile procedures and with 1% lidocaine local anaesthetic in 6 PD and 6 control subjects (Table 1) at the Parkinson's Disease Research Clinic at the John Van Geest Centre for Brain Repair, University of Cambridge/Cambridge University Hospitals NHS Trust, UK. CSF (5 ml) was collected and centrifuged for 10 min at 300 g at 4 °C. The supernatant was collected and stored at −80 °C. CSF from 20 AD and 20 control subjects with matched serum samples was collected at the highly coordinated memory outpatient clinics in Sweden following a standardized protocol[41].

**Extracting soluble aggregates from brain tissue**

Post-mortem brain tissue from the amygdala in 9 PD donors and 5 controls, with no known history of neurological or neuropsychiatric symptoms, was acquired from the Cambridge Brain Bank. The post-mortem work was approved by the London−Bloomsbury Research Ethics Committee; 16/LO/0508. The brain samples have been

voluntarily donated without any compensation and with the ethics allowing the release of associated anonymised, non-identifiable clinical and pathological metadata. Demographic and clinical characteristics of donors are shown in Table 3. Brain tissue was flash-frozen and stored at −80 °C. For extraction of soluble aggregates, a previous published protocol was adapted[32]. 300 mg tissue samples were cut into smaller pieces and placed into an Eppendorf containing 1.5 mL of artificial cerebrospinal fluid buffer (aCSF, 124 mM NaCl, 2.8 mM KCl, 1.25 mM NaH2PO4, 26 mM NaHCO3; pH 7.4, supplemented with 5mMEDTA, 1mMEGTA, 5 µg/mL leupeptin, 5 µg/mL aprotinin, 2 µg/mL pepstatin, 20 µg/mL Pefabloc, 5mMNaF) for 30 min at 4 °C. Afterwards the samples were centrifuged at 2000 × g for 10 min and the upper 90% of the supernatant was transferred into a fresh tube. This solution was then centrifuged at 14,000 × g for 2 h and again the upper 90% supernatant was collected and dialysed for 72 h using Slide-A-Lyzer cassettes (MKCO 2 kDa, Thermo Scientific, Cat. 66330) with three buffer exchanges against aCSF buffer at 4 °C. The samples were aliquoted and stored at −80 °C and each aliquot was used for just one experiment to avoid unnecessary freezing/thawing cycles.

**Preparation of inflammasome-activated THP-1 cell samples**

The immortalized human monocyte cell line THP-1 from American Type Culture Collection was maintained in RPMI 1640 (Invitrogen) containing 10% HI-FBS, 2mM L-Glutamine, and 1% penicillin and streptomycin. Monocytes were differentiated into macrophages with 10 nM Phorbol 12-myristate 13-acetate (PMA) at a density of 21052.6/ cm² in a 6 well plate for 24 h. Macrophages were then recovered in fresh complete media for another 24 h. Ultra pure lipopolysaccharide from E. coli 0111:B4 at 200 ng/ml was added to all macrophages for 3 h to prime the cells. Nigericin was added at 10 µM and incubated for increasing time points before conditioned media was collected and cells lysed in 50 µl per well NP-40 buffer (150 mM NaCl, 50 mM Tris−HCl pH 7.4, 1 mM EDTA, 1 % NP-40). All samples were snap frozen in dry ice before SiMPull analysis. Conditioned media was also assayed for IL-1β by ELISA to confirm inflammasome activation using the DuoSet ELISA kit (Supplementary Fig. 1).

**Coverslip preparation**

Coverslips were coated according to our recently-published protocol[30]. Each glass coverslip (26 × 76 mm, thickness #1.5, VWR, Cat. No. MENZBC026076AC40) was plasma cleaned for 15 min. 50-well PDMS chamber gasket (cut from a CultureWell chambered coverglass, Sigma, cat. no. GBL103350-20EA) was attached to the cleaned coverslip. To prepare a non-sticky glass surface, we exploited a hydrophobic

self-assembling monolayer approach using a RAIN-X Rain Repellent solution. Rain-X and isopropanol were mixed in 1:1 proportion, 0.02 μm filtered and added to each well (10 μl). The coated coverslip was left at room temperature until full evaporation of liquid. The wells were then washed twice with 10 μl of PBS by pipetting the liquid in and out to remove RAIN-X residuals. Coverslips were prepared freshly and used in the same day as a platform for the SiMPull assay.

## ASC speck SimPull assay

A Rain-X hydrophobic glass coverslip was prepared according to the previous section. Then, 10 μl of 0.1 mg/ml Neutravidin (Thermo Scientific, 31000) was added to each well (last 4 lines, 40 wells and first line were unused due to its closeness to the edge) for 10 min and then washed twice with PBS by pipetting the liquid in and out. To minimise unspecific binding, the wells were blocked with 1% filtered solution of pluronic f127 diluted in PBS for 60 min followed by washing with PBS twice and blocking again with 3 mg/ml of BSA (Molecular Biology Grade, New England Biolabs) in PBS-T (0.05% Tween-20 in PBS) for 30 min. 10 μl of 10 nM anti-ASC antibody in PBS-T was added to each well and incubated for 15 min and then washed off the unbound Ab with PBS-T twice. Then, 10 μl of the undiluted sample (or 3-fold diluted in PBS for serum and THP-1 cell lysates only) was added to each well for 90 min and then washed twice with PBS-T. For detection of captured ASC specks, 10 μl of 5 nM detection anti-ASC antibody in PBS-T was added to the well for 15 min and washed with PBS-T twice. Finally, each well was covered with 10 ul of PBS for diffraction-limited imaging. For reliability of results, 2 quality control wells were included in each experiment: (1) buffer control, when PBS was added instead of sample and (2) no capture control, when there was no capture antibody but the correct detection antibody, to check the amount of non-specific absorption of sample onto the imaging surface. The amount of α-syn, Aβ and p-tau aggregates in the AD cohorts of serum samples were measured using the same SiMPull assay protocol as for ASC specks with the same protein-specific antibody for capture and detection (see Antibody Section for details). The amount of α-syn and Aβ aggregates in the PD serum cohorts were measured using our recently-published hybrid aptamer-antibody SiMPull assay (APSiMPull)[29] which specifically detect aggregated forms of α-syn and Aβ in serum. For this, we used a T-SO508 aptamer which binds both α-syn and Aβ aggregates with common β-sheet intermolecular structure as the capture probe and for detection of aggregates of a specific protein we used an antibody (either 6E10 for Aβ or syn-211 for α-synuclein). For $d$STORM imaging, another 3 layers of PDMS chambers were stacked onto the coverslip to increase the well capacity. 16.5 μl $d$STORM buffer (50 mM PBS-Tris, 0.5 mM glucose, 1.3 μM glucose oxidase, 2.2 μM catalase, and 50 mM mercaptoethylamine (MEA)) was then added to each well. To maintain the pH during imaging, the top of the chambered coverslip was sealed using a second cleaned coverslip. To further reduce oxygen penetration, the edges of the integrated coverslip complex were further coated with nail polish and parafilm.

## Workflow of SiMPull method

Our ASC speck SiMPull is a three-step assay (see Fig. 1A). (1) Assay step 1 (see the section above): preparation of a functional surface containing multiple single ASC aggregates directly immunoprecipitated from a sample. This step is carried out using a 40-well CultureWell™ gasket coverslip with a 10 μl well capacity. High-throughput assay performance was achieved using a 12-chanel Integra VOYAGER electronic pipette and electronic repeater Eppendorf pipette which allow to automatically prepare the surface coating, perform washing steps, add the surface capture antibodies, add the samples and then add the imaging antibodies. A coverslip preparation with 40 different samples/wells ready for imaging takes 3 h. (2) *Assay readout 2:* the measurements of the amount of single ASC aggregates present in each sample well. This step

is performed using a total internal reflection fluorescence (TIRF) microscope (see the Imaging section for details) and takes 1 h for a 40-well coverslip. During this step we count the number of diffraction-limited spots (each representing a single aggregate) per field of view which allow us to measure the levels of ASC protein aggregates in each analysed sample. Because the size of ASC aggregates in human biofluids are close to the diffraction limit of light (~220 nm for our optical set-up) we super-resolve them to accurately measure their size and shape in step 3. (3) Assay readout 3: the measurements of the size and shape of individual ASC aggregates in each sample well. After the second step, the $d$STORM buffer is added to each well and super-resolution imaging of the aggregates is performed. This step is carried out *o*vernight (15 h) using automated data collection.

## Antibodies and aptamer

Unconjugated rabbit anti-ASC (clone AL177, 1 mg/ml) antibody against aa at the N-terminal human ASC was purchased from AdipoGen. The Alexa 647 conjugated anti-ASC and AT8 antibodies were generated using Zip Alexa Fluor™ Rapid Antibody Labeling Kit (Cat. No. Z11235) and the biotinylated anti-ASC and 211 antibodies were generated using a Pierce™ FITC Antibody Labeling Kit (Cat. No. 53027). The biotinylated 6E10 antibody was purchased from Biolegend (Cat. No. 803007). The biotinylated p-tau (Ser202, Thr205) antibody (AT8) antibody was purchased from ThermoFisher (Cat. No. MN1020B). Alexa 647 anti-Aβ antibody (6E10) targeting Aβ amino acids 1-16 was purchased from Biolegend (Cat. No. 803021 and 803013). Unconjugated (Cat. No. sc-12767) and Alexa 647 conjugated (Cat. No. sc-12767 AF647) anti-α-syn (211) antibodies were purchased from Santa Cruz biotech which recognise amino acid 121-125 of human α-synuclein. Alexa 647 Mouse IgG1 (Cat. No. MA5-18168; PRID: AB_2539542, Invitrogen), Alexa 647 Rabbit IgG (Cat. No. 3452S; RRID: AB_10695811, Cell Signalling Technology), Bovine Serum Albumin (B9000S, New England biolabs) were also purchased commercially. Biotinylated T-SO508 aptamer (GCCTGTGGTGTTGGGGCGGGTGCG) was purchased from ATDBio (Southampton, UK) and purified by high-performance liquid chromatography (HPLC). The aptamer recognises β-sheet structure and is specific to both α-syn and Aβ oligomers[42].

## Immunoprecipitation

Immunodepletion of ASC protein from serum ($n = 2$ PD samples) was performed using our previously published immunoprecipitation (IP) protocol[14]. In brief, 5 μg of rabbit anti-ASC (AL177) antibody (1:50, 1 mg/mL) or rabbit IgG isotype control non-target antibody was incubated with 1.5 mg magnetic Dynabeads® protein G (30 mg/mL, Invitrogen) in a total volume of 200 μL 0.02% Tween-20 (PBST) in separate 1.5 mL Protein LoBind Tubes (Eppendorf AG) on a rotating wheel at 4 °C for 2 h. The beads were washed with 500 μL PBST three times. Serum sample (200 μL) was added to the beads tube and incubated overnight at 4 °C while rotating. The supernatant was collected into the 0.5 mL eppendorfs and kept on ice. The samples were analysed immediately in the ASC SiMPull assay.

## Aggregate stability assay for ASC specks

Prior adding to SiMPull assay, samples underwent a denaturation treatment with different concentrations of guanidine hydrochloride (Gdn HCl) which affects the secondary and tertiary structure of protein aggregates without affecting the primary structure of monomers. First we optimised our protocol to determine the molar concentration of Gdn HCl at which the quantity of ASC speck aggregates decreased in value to $1/e \approx 0.4$ relative to the untreated sample. For this, 10 μl aliquots of sample (undiluted PD brain homogenate or PD serum) were individually treated with decreasing concentrations of 8 M Gdn HCl diluted in TRIS-buffered saline (TBS), pH 8 (Thermo Scientific) so that the final concentration in the sample varied from 0.8 M to 4 M with increment 0.5 M. After 60 min

incubation at room temperature, individual samples were diluted back to the diminished concentrations of Gdn HCl in TRIS-buffered saline (TBS), pH 7.4 at the final concentration of 0.4 M in all samples and immediately added to the SiMPull coverslip for ASC speck detection. For all of diseased serum, brain homogenate and CSF samples there was an exponential step decrease in the number of ASC specks compared to untreated sample with 0.8 M Gdn HCl treatment but not for the CSF sample without neurological disease (<30% decrease). To measure the range 0.72 M–0.05 M Gdn HCl with a finer increment, 10 µl aliquots of sample (3-fold diluted PD brain homogenate or undiluted PD serum and AD/HC CSF) were individually treated with 1 µl of decreasing concentrations of 8 M Gdn HCl diluted in TRIS-buffered saline (TBS), pH 8 (Thermo Scientific) so that the final concentration in the sample varied from 0.72 M to 0.05 M. After 60 min incubation at room temperature, each condition sample was immediately added to the SiMPull coverslip for ASC speck analysis.

## Imaging

Imaging was performed using a custom-build total internal reflection fluorescence (TIRF) microscope. A Nikon Ti2 Eclipse inverted microscope is integrated with 100×1.49 NA oil-immersion objective (UPLSAPO, 100x, TIRF, Olympus) and a perfect focus system. An excitation laser beam (Oxxius, 635 nm) was circularly polarised by a quarter-wave plate (WPQ05M-405, Thorlabs) and focused onto the back focal plane of the objective. The fluorescence emission was collected using the same objective and separated by a dichroic beamsplitter (Di01-R405/488/561/635, Semrock), with filtering performed by a long-pass emitter (BLP01-635R-25, Laser 2000). Emission is imaged onto an air-cooled EMCCD camera (Photometrics Evolve, EVO-512-M-FW-16-AC-110) with frame transfer and pre-exposure mode (electron-multiplying Gain of 11.5 e-1/ADU and 250 ADU/photon). The camera was operated with pre-exposure non-overlap mode to precisely control the exposure time. The open-source software Micro-Manager 1.4 was employed to automate image acquisition. 638 nm laser (Cobolt MLD 638, Cobalt) was used to excite Alexa 647 dyes. For diffraction-limited imaging, 1.5 mW of laser power was applied and images were acquired with an exposure time of 50 ms and frame number of 50. For $d$STORM imaging, 150 mW of laser power was applied and images were acquired with a camera exposure time of 30 ms and frame number of 8000. The images were recorded at frame rate of 10 M Hz 16 bit (33.7 ms per frame) resulting in total acquisition time of ~9 min for a video. Alternatively, the use of the overlap mode (simultaneous exposure-readout) which ensured continuous imaging (100% duty cycle) gave similar results on the size and shape distributions of ASC specks (within 95% confidence interval) and similar average precision of 13 ± 5 nm compared to the non-overlap mode (Supplementary Fig. 11). Continuous illumination by 405 nm laser (LBX-405-50-CIR-PP, Oxxius) at 10 mW (at the source) was applied for active photoswitching. The use of continuous 405 nm laser did not led to faster photobleaching in our system (see comparison of the number of localisations as a function of time when measured with versus without the UV laser in the Supplementary Fig. 6E). Average number of localisations per ASC aggregate in a $d$STORM image was 58 (see Supplementary Fig. 10F). The pixel size of the camera was measured 103.5 nm.

## ELISA quantification of IL-1β concentrations

Quantitative determination of IL-1β for the THP-1 conditioned media was performed using a human IL-1β DuoSet ELISA kit. Assay signal was measured on a CLARIO 430-1262 ACU plate reader. Assay range is 3.9–250 pg/mL. An assay signal was measured at 450 nm wavelength. Signals at 450 nm were subtracted from plastic background measured at 570 nm wavelength. The assay was performed according to the

manufacturer's instructions. Briefly, 100 µl of the capture antibody diluted in PBS was added per well and the sealed plate was incubated overnight at 4°C. Wells were washed 3 times with the wash buffer (0.05% Tween® 20 in PBS, pH 7.2–7.4). Then, plates were blocked with 1% BSA in PBS, pH 7.2-7.4, 0.2 µm filtered for 1 h and washed with the wash buffer 3 times. The media samples were run undiluted. Subsequently, 100 µl of undiluted samples and dilution series of human IL-1β standard were incubated for 2 h at room temperature, washed with the wash buffer 3 times and incubated with the 75 ng/ml detection antibody diluted in the reagent buffer (0.1% BSA in PBS, pH 7.2–7.4) for 2 h at room temperature. Wells were washed 3 times with the wash buffer. 100 µl of streptavidin-HRP solution diluted in the reagent buffer was added to each well and the plate was incubated for 20 min at room temperature in the dark followed by 3 times washing with the wash buffer. Then, 100 µl of the substrate mixture (1:1 stabilized hydrogen peroxide + stabilized tetramethylbenzidine) was added to each well and the plate was incubated for 20 min at room temperature in the dark. Finally, 50 µL of stop solution (2 N sulphuric acid) was added to each well, the plate was gently tapped to ensure mixing within the wells and measured immediately in a reader.

## Ultra-sensitive quantification of pro-inflammatory cytokine panel and CRP levels in serum

In AD/control and PD/control samples, a panel of serum cytokines (IFN-γ, IL-1β, IL-2, IL-4, IL-6, IL-10, IL-12p70, IL-13, TNF-α) were measured using the MesoScale Discovery (MSD) S-plex proinflammatory panel 1 kit. Serum c-reactive protein (CRP) was measured using the high-sensitivity Siemens Dimension EXL autoanalyser. Serum samples were run undiluted. The MSD assays were performed in duplicate and the calculated mean was used for results. The CRP analysis was performed in singleton samples.

## Analytical precision of the ASC speck SiMPull assay

Inter-assay precision was calculated by measuring the 20 serum samples and 3 ASC standards (3.75 min, 30 min and 60 min nigericin media from inflammasome-activated THP-1 cells) on two different plates to monitor plate-to-plate variation. Intra-assay precision was calculated from the duplicate measurements of 1 AD and 1 non-AD serum samples within one plate. CVs (coefficient of variation) of < 20% was considered acceptable for precision. For both the ASC standards (conditioned media) and serum samples, average inter-assay precision was 14% in the media and 15% in the 20 serum samples. Average intra-assay precision was also acceptable with 12% in the media and 7% in the serum samples (Supplementary Table 1).

## Data analysis

Data were analysed using MATLAB (R2020b). The diffraction-limited data were analysed using in-house software called Path-Connected Aggregate Recognition (PCAR) which is freely available from https://github.com/LobanovaEG-LobanovSV/PCAR. $d$STORM data were analysed using established ImageJ plug-ins. The drift correction, image reconstruction, and morphology analysis was performed by mean shift algorithm[43], ThunderSTORM (version 1.3)[44], and morphology library[45], respectively. A custom-written Matlab code was used to integrate mentioned plug-ins and automate data analysing. The code is available from https://github.com/YPZ858/Super-res-code/issues/1. The lateral localisation uncertainty in our $d$STORM images was 17 ± 7 nm, with > 95% of single molecules having moderate resolution < 30 nm. This indicated a low risk of detecting multiple emitters in the same frame (overlapping point spread functions). The cumulative histograms of the area, perimeter and circularity distributions and their relative differences are generated to investigate the morphological differences in the ASC specks between diagnostic groups. To relate area and perimeter to the size of ASC speck, the size was estimated as a circle with a

diameter determined as two square roots of the area divided by π ($d = 2\sqrt{A/\pi}$) or the perimeter divided by π ($d = P/\pi$).

## Statistics and reproducibility

All experiments were performed in three independent replicates. For normally distributed data, two-tailed unpaired $t$-test was employed. Otherwise, the permutation (exact) test was used except for the sample size, for which the binomial test was employed. Statistical significance was indicated when $p < 0.05$. $P$-values are indicated by $*p < 0.05$; $**p < 0.01$; $***p < 0.001$; $****p < 0.0001$; $*****p < 0.00001$; In our previous studies 10 diseased by 10 control serum samples were sufficient to observe statistically significant differences ($p = 4.3 \times 10^{-5}$) in the levels of studied protein aggregates using our single-molecule imaging technique giving a large effect size (Cohen's d) of 2[14]. Therefore, for validation of developed biomarkers here the minimum sample size $n = 10$ per group was used. This analysis allowed to achieve power of 99% for detecting case-control differences at a significance level of $p = 0.05$. Diagnostic performance of each candidate biomarker was assessed using Receiver operating characteristic (ROC) curve analysis and its accuracy was evaluated by area under the curve (AUC). To identify the most promising biomarker combination, we ran a loop in MATLAB which calculates various simple combinations of measured biomarkers and their AUCs. We then ranked the output combinations of biomarkers based on their AUC values from most (highest AUC) to least (lowest AUC) promising. For the combined biomarkers comprising the ratio of the number of ASC specks to the number of Aβ aggregates $ASC/A\beta$ and the ratio of the sum of the number of ASC specks and the number of p-tau aggregates to the number of Aβ aggregates $(p\text{-}tau\text{-}AT8 + ASC)/A\beta$ used to distinguish early AD from control serum and AD dementia from control serum, we used the normalised number of Aβ aggregates. The normalisation factor was determined as a maximum number of Aβ aggregates within individual diagnostic group (AD, AD dementia and control). The relationship between the number of ASC specks in serum versus CSF from the same individual subjects and the number of ASC specks with the cytokine levels in serum were assessed using the Pearson's correlation analysis or the Spearman Rank Correlation when the data dependence was non-linear. The correlation ($R$) was considered to be statistically significant when $p < 0.05$.

## Reporting summary

Further information on research design is available in the Nature Portfolio Reporting Summary linked to this article.

# Data availability

Source data are provided with this paper. All other data are available from the corresponding author on request. Source data are provided with this paper.

# Code availability

The code for the diffraction-limited data analysis is publicly available at https://github.com/LobanovaEG-LobanovSV/PCAR. The code for the super-resolution data analysis is publicly available at https://github.com/YPZ858/Super-res-code/issues/1.

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

## Acknowledgements

Human post-mortem brain tissue was acquired from the Cambridge Brain Bank (Cambridge University Hospitals). The Cambridge Brain Bank is supported by the NIHR Cambridge Biomedical Research Centre (NIHR203312). We gratefully acknowledge the participation of all our patient and control volunteers. We greatly thank the NIHR Core Biochemistry Assay Laboratory at Cambridge Biomedical Research Centre for performing the serum cytokine assays for AD/control samples. We also greatly thank Dr John S.H. Danial for very helpful discussions on super-resolution microscopy. A small amount of in-house biotinylated ASC antibody and dye-labelled AT8 antibody were supported by Jeff Y.L. Lam and Dorothea Böken. The work was supported by a grant from Parkinson's UK (G-1901), the UK Dementia Research Institute (which receives its funding from UK DRI Ltd), the UK Medical Research Council, Alzheimer's Society and Alzheimer's Research UK (ARUK-PG2020A-009), ARUK-PG2020A-009 and the Royal Society. CHWG was supported by the Medical Research Council (MR/R007446/1 and MR/W029235/1), and the NIHR Cambridge Biomedical Research Centre (NIHR203312). JBR is supported by the NIHR Cambridge Biomedical Research Centre (NIHR203312) and Wellcome Trust (103838; 220258). MM is supported by Race Against Dementia Alzheimer's Research UK (ARUK-RADF2021A-010) and the NIHR Cambridge Biomedical Research Centre (NIHR203312). HZ is a Wallenberg Scholar and a Distinguished Professor at the Swedish Research Council supported by grants from the Swedish Research Council (#2023-00356; #2022-01018 and #2019-02397), the European Union's Horizon Europe research and innovation programme under grant agreement No 101053962, Swedish State Support for Clinical Research (#ALFGBG-71320), the National Institute for Health and Care Research University College London Hospitals Biomedical Research Centre, and the UK Dementia Research Institute at UCL (UKDRI-1003). The views expressed are those of the authors and not necessarily those of the NHS, the NIHR or the Department of Health.

## Author contributions

E.L., Y.P.Z., D.E., J.B., L.K., M.M and A.Q. performed and analysed the experiments. D.K., C.E.B, C.H.W.-G., J.B.R., H.Z., M.T. and K.T. supervised the project. E.L., Y.P.Z., C.E.B and D.K. conceived the idea, designed the study and wrote the manuscript. All authors discussed the results and proofread the manuscript.

## Competing interests

The authors of the paper, E.L., Y.P.Z. and D.E., are inventors in the international patent application (application number: GB2317286.9, status: pending, applicant: Cambridge Enterprise Limited). The patent application has been filed on the methods of protein aggregate detection for diagnosis of neurodegenerative diseases including a method for detecting ASC specks in human biofluids, which is described in this manuscript. All other authors declare no competing interests.
