## [Transparent Peer Review file · Nature Communications]

ASC specks as a single-molecule fluid biomarker of inflammation in neurodegenerative diseases

Corresponding Author: Dr Evgeniia Lobanova

Version 0:

Reviewer comments:

Reviewer #1

(Remarks to the Author)

This is a manuscript by a very knowledgeable group to study several measurements (quantity, morphology) of individual extracellular inflammasome (ASC specks) as a biomarker for inflammation in serum, cerebrospinal fluid and brain samples from patients with Parkinson's or Alzheimer's disease.

They first analyzed ASC speck aggregates generated in THP-1 cells and showed that ASC specks form early in the inflammatory response. They then moved into biomarker evaluation of ASC speck numbers, which is the main part of the manuscript. The authors state "...we introduce a new assay that has potential as an ultrasensitive biomarker of inflammation in the blood of AD and PD...":

With sample sizes of CSF (6 PD and 6 HC) and serum (divided in two cohorts: 10 PD and 10 HC as well as 8 PD and 5 HC plus 10 AD vs 10 HC) and soaking brain tissue (9 PD, 5HC and 6 AD) I am not sure if it makes sense to present ROC curves and present this assay as a biomarker of inflammation. I am wondering if the assay is so complicated that larger cohorts are anyway not feasible to analyze? How can this assay be developed for high throughput at all?

Authors state several times that they analyzed the samples in EARLY PD and AD, but the MDS-UPDRS numbers and especially also the MMSE mean of 17 in AD is advanced dementia (which can barely be after a mean of 1.6 years of dementia)-something in these cohorts is weird and needs a deeper look.

The authors next performed some control experiments indicating aggregate specificity and assay stability including conformation stability. Here, I think depletion experiments are lacking showing the depletion of the assayed signal.

For more specificity the authors added to the number of ASC specks (and later also morphologic aspects) the numbers of aggregates of beta-amyloid, tau and alpha-synuclein for calculating ratios to the ASC specks as "composite biomarker". Here it would be important to show the specific markers alone. Also established SAA methods would be good to add. I am also lacking inflammatory marker protein(s) here. Authors do state in the method section, that the conditioned cell media was also assayed for IL-1beta by ELISA to confirm inflammasome activation. Such a quantification should also be carried out in the biofluids to correlate with the ASC Speck numbers and morphology.

Overall, I think this is an interesting assay and way to visualize and analyze inflammasomes. I am not convinced about the potential for such an assay as a robust biomarker for larger cohorts, rather could this assay be utilized to detect inflammation response very early in the disease. Existing cohorts of prodromal subjects and with different disease stages would be necessary here.

Authors state at the end of the discussion that their assay might be useful as a biomarker to detect target engagement of inflammasome inhibitors in clinical trials-I think this cannot be stated with the data shown as neither performed the authors inflammation inhibition nor enough data to claim this as a biomarker.

Minor:

some abbreviations of the figures are not included in the legends.

How was the neuropathological diagnosis of case PD8 been done if Lewy Body Braak staging is missing? Knowing of co-aggregation and co-pathology all staging should be done for all brains.

Reviewer #2

(Remarks to the Author)

The authors present a novel smart and easy single-molecule pull-down assay (combined with dSTORM) characterizing

specific crucial parameters of individual ASC specks in serum, CSF and brain from PD and AD patients. The authors achieved greater biomarker performance and accuracy by combining the quantity of ACS specks with other biomarkers of protein aggregate diseases. This novel approach has the potential to distinguish healthy people from people at early stages of aggregate diseases (PD & AD) as well as to potentially characterize specific therapeutic treatments at the single-molecule level in clinical trials.

The manuscript is well written and most of the illustrations are clear to understand. Overall, this work is a valuable contribution to the field. However, I would have some crucial questions which need to be addressed prior publishing the paper in Nature Communications.

Major points:

1. Regarding the coverslip preparation, the authors refer to reference #39 which can not be found online (maybe also in review or recently submitted? [not stated]) (see also minor point #3). Furthermore, the authors did plasma cleaning and afterwards a hydrophobic self-assembling monolayer approach using Rain-X. Afterwards, this prepared surface is treated with „biotinylated NeutrAvidin“ but where does the biotin binds to? It is clear, that the authors use NeutrAvidin to immobilize the biotinylated capture antibody but it is not clear why the NeutrAvidin must be biotinylated? (see Fig. 1A). Or is this an error in the illustration? There is also no protein on the surface or at least stated in the manuscript that will bind to the biotinylated NeutrAvidin to immobilize it. The authors also blocked the treated surface with BSA, but wouldn't it be obvious to only use a mixture of BSA and biotinylated BSA or even only biotinylated BSA to assure the binding of NeutrAvidin, building up the SiMPull assay?
2. The authors did certain control measurements including SiMPull assay without capture antibody as well as SiMPull with capture antibody but with a non-target Alexa647 IgG which sounds sensible. But to be sure, concerning unspecific binding, I would prefer to also test the SiMPull assay without the capture antibody but with the correct target Alexa647 IgG in buffer only, THP-1 cell lysate and THP-1 cell media to validate unspecific binding of the detecting antibody (see also minor point #11).
3. In Fig. 1B+E the authors showed box plots of the SiMPull assays measured. „The dots in B and E represent independent replicates $n=3$ “. But for this sample size ($n=3$) it is hard to generate adequate statistics. For these crucial experiments of the SiMPull assay, I prefer to increase the sample size to at least $n = 5$ or even better $n = 7-10$ independent measurements.
4. The dSTORM measurements were done with an exposure time of 15 ms (66.67 Hz) for 2 min (8000 frames), with a continuous UV illumination (405 nm). For high quality dSTORM measurements an appropriate switching behaviour of the fluorophores used has to be reached. The individual fluorophores should show long non-fluorescent „off-states“ (several seconds) and short fluorescent „on-states“ (several milliseconds). At the beginning of an experiment, most fluorophores should be transferred into the off-state by irradiating the whole sample in the FOV with the appropriate laser wavelength and power (if measurements are done in the TIRF mode, ensure that the fluorophores show an appropriate switching behaviour, which can easily be achieved by changing the illumination mode to EPI for a short period of time and then go back to the TIRF mode for measurement). This „off-state“ can be elongated using oxygen scavenger systems like the system the authors used. This ensures, that the triplet state as well as the „off-state“ is not quenched by molecular oxygen. Adequate off-state times reduce the risk of overlapping signals (overlapping point spread functions (PSF)) from neighboring fluorophores in the diffraction-limited space, minimizing the risk of (blinking) artifacts and boosting the quality of the localization data and high resolved images. The on-state of the fluorophores used (Alexa647) can also be repopulated by UV light. This could also increase the probability of multiple fluorophores in the on-state within the diffraction-limited space. Which could lead to high amounts of artifacts, false localizations and bad data quality. The quality of the images shown varies greatly and the images reflect the localization data. Maybe the authors can also upload a dSTORM movie to have a look at the switching behaviour. As stated previously, the authors measured for 2 min (8000 fr), which seems to be a very short measuring time. Maybe because of that, the authors used continuous UV illumination to detect a high amount of emitters within this short measuring time. But this could adversely affect the quality of the measured dSTORM localization data. But this is a crucial point for the whole paper and method. To improve the quality of the images as well as the localization data, I would recommend that the authors also do some measurements of the SiMPull assay for the THP-1 cell lysate and media without continuous UV illumination (like for Fig. 1 B and E) for 5 – 10 min (afterwards the movie can also be shortened to 2 min if there is no further positive effect on the localization data quality after to 2min) to minimize the risk of bad blinking behaviour and overlapping PSF (within this short measuring time) improving the quality of the data.

Minor points:

- 1) The authors are doing direct stochastic optical reconstruction microscopy (dSTORM) but cite a publication about „STORM“ using FRET via Cyanine3 and Cyanine5 to achieve blinking of Cy5. So I would recommend to cite a dSTORM paper where Cyanine dyes are directly photochemically switched between on- and off-state. (This refers to reference #13: Rust, M. J., Bates, M. & Zhuang, X. Sub-diffraction-limit imaging by stochastic optical reconstruction microscopy (STORM). Nat Methods 3, 793–795 (2006))
- 2) It is common practice to italicize the letter d in dSTORM, so I would recommend to change all d in dSTORM to italicized letters. And I would also recommend to italicize the word direct when writing out dSTORM (page 3). Furthermore, the authors often state STORM instead of dSTORM, so please change STORM to dSTORM.
- 3) Concerning some references, it is not possible to find the reference #29 (Zhang, Y. P. et al. Imaging protein aggregates in

Parkinson's disease serum using aptamer-assisted single-molecule pull-down. Chem Sci (2023)) as well as #39 (Zhang, Y. P. et al. A simplified imaging surface for versatile and high-throughput single-molecule studies. Anal Chem (2023)). The authors have stated the journal but without any additional specific numbers like edition or pages. Are these papers already published, in a review process or even recently submitted and the authors forgot to mention?

4) Beside the previous remark, could the authors also complete reference #6 (King, E. et al. Inflammation in Mild Cognitive Impairment due to Parkinson's disease, Lewy Body disease and Alzheimer's disease) ,... Int J Geriatr Psychiatry. 2019 Aug;34(8):1244-1250. doi: 10.1002/gps.5124. Epub 2019 May 16. PMID: 30993722" is missing

5) On page 4 it is stated: „The SiMPull assay is capable of detecting ASC specks in both the NLRP3 inflammasome-activated THP-1 cell lysates (LPS-primed nigericin-activated) and those secreted into the media compared to untreated negative controls (Fig. 1C)“. The authors refer to Fig. 1C, but doesn't refer this sentence to Fig. 1B/E or Fig. 1C/D or at least to Fig. 1B-E and not only to Fig. 1C?

6) Regarding Fig. 1A: Could the authors make the caption a little bit more detailed, for example what the yellow dots are or belong to? So that at the end it is easier to follow only by looking at the illustration of the SiMPull assay.

7) In Fig. 1A please correct „biotinylated ASC (AL117) antibody“ to antibody.

8) In Fig.1, B and E is missing.

9) Concerning Figure 1: For clearer illustration I would recommend rearranging the order of the images and graphs.

a. Top row: A) ASC SiMPull (left), B + C) THP-1 cell lysates and cell media (dSTORM-pictures) (right)

b. Middle row: D) THP1-cell lysates box plot (left), E) media box plot (right) if possible with the same y-axis scale to directly see the differences

c. Lower row: F and G (aggregate area and circularity)

10) I would also recommend to change the way of illustration in Fig. 1 F and G, because it is very confusing to correctly assign the geometric shapes. I would prefer to use clearly distinguishable shapes or even different colours. (The design of the other box plots like Figure 2 etc. is way easier since there is only a distinction between two groups HC and PD or AD)

11) Which degree of labeling (DOL) did the authors use for the target Alexa647 antibody? Because it is also well known, that higher DOL (>>4) could also lead to a high amount of unspecific binding of the antibodies.

12) In the method section under „ASC speck SimPull assay“, in the second sentence the authors mentioned the word „twice“ twice, so please delete one twice („...washed twice with PBS twice...“).

Reviewer #3

(Remarks to the Author)

In the article by Lobanova et al, the authors describe the use of ASC inflammasome specks as a sensitive fluid biomarker of neuroinflammatory disease severity in Parkinson's and Alzheimer's disease. The authors claim that the increased quantity and abnormal morphology of ASC specks in the blood of patients makes them a candidate biomarker of inflammation for Parkinson's and Alzheimer's diseases. When combined with other measurements the diagnostic accuracy can be further improved.

The article is clearly written and the study was well designed and implemented. The conclusions are supported by the results to a large extent. However some issues need

Since the study is based on measuring inflammation as a biomarker of PD/AD it should be considered to include some of the more common markers of inflammation in the analysis (C-reactive protein, IL-1 β , IL-6, INF- γ , MCP-1, TNF- α , erythrocyte sedimentation rate and plasma viscosity) to see how they correlate with ASC specks. And to see if ASC specks present the added value for the diagnosis.

Also for CSF and plasma samples, were patients free of any disease that can cause even minor inflammation? Was white blood cell count performed on CSF and blood?

Since the study is based on inflammation markers, shouldn't the AD case with Burkitt's Lymphoma and pneumonia be excluded from the analysis?

In addition, when it is not stated, what was the death cause? Was it always natural, or some other death causes were present?

The CSF collection site is not specified. It is well known that protein gradients exist in CSF and differences in lumbar puncture sites, and even patient height, will affect the results of any assay. To omit the problem the measured values should be normalized to for example the total protein level.

Another aspect is if the CSF was tested for contamination with blood. Some measurements will be strongly affected by even a small amount of blood contamination.

Finally, the discussion lacks the section about the limitations of the study. For example, authors claim that their method can

be used as a biomarker, however how easy will it be to implement it in routine clinics? How long the test will take? What will be the cost of analysing the sample?

Minor issues

No post-mortem interval is specified for AD patients.

Figure 1 – Only A, F and G captions are visible

Scale bars on the figures should start from 0

The same scale bars should be used when comparing the same thing in 2 cohorts. E.g. Figure 2 A and B.

Version 1:

Reviewer comments:

Reviewer #1

(Remarks to the Author)

The manuscript has definitively improved with the new independent samples, the additional inflammatory values and methodological aspects. I still see quite some incongruency in the cohorts (quite high UPDRS numbers of short duration etc.) and despite the comments by one other reviewer there are a significant number of references that are not complete.

Reviewer #2

(Remarks to the Author)

I would like to thank the authors very much for addressing and revising most of the points and comments mentioned. However, there are some crucial points that have still not been addressed.

1) The authors performed dSTORM (direct Stochastic Optical Reconstruction Microscopy) measurements, but they are still referring to STORM (Stochastic Optical Reconstruction Microscopy), which has some differences from dSTORM. Please modify the respective reference (#13) accordingly and cite a dSTORM paper (e.g., M. Heilemann, S. van de Linde, or M. Sauer), for example M. Heilemann, S. van de Linde, M. Schüttelz, R. Kasper, B. Seefeldt, A. Mukherjee, P. Tinnefeld, and M. Sauer, "Subdiffraction-resolution fluorescence imaging with conventional fluorescent probes," *Angewandte Chemie International Edition*, vol. 47, no. 33, pp. 6172-6176, 2008"

2) The authors are still writing STORM instead of dSTORM. Please correct it to dSTORM in the main text as well as in the supplementary information.

a. main text

i. In line 617 of the revised manuscript

ii. In line 687

iii. Figure caption 5. Line 972

b. Supplementary information

i. Figure caption 4, line 2

ii. Figure caption 5, line 1

3) Please also italicize the letter d in dSTORM in the entire manuscript

4) The authors have not really answered the major point 4 regarding the performance of the dSTORM measurements.

a. Why do the authors use continuous UV illumination? On the one hand, the on-state of the fluorophores used (Alexa647) can be repopulated by UV light. This could increase the probability of multiple fluorophores in the on-state within the diffraction-limited space, especially if the labeling density of the target structure is high. This could lead to high number of artifacts, false localizations and bad data quality. The quality of the images shown varies greatly and the images reflect the localization data. On the other hand, continuous UV illumination leads to a higher risk of faster photobleaching of the dyes, resulting in a rapid loss of emitters and information, especially with a high irradiation power of 10 mW like the authors stated in the methods. To minimize the aspects stated no or pulsed UV illumination would be better. Could the authors also do some simple control measurements without UV illumination? Since I recommended to do some measurements of the SiMPull assay without continuous UV illumination to minimize the risk of bad blinking behavior and overlapping PSF improving the quality of the data.

b. The authors mentioned that they used 15 ms exposure time for imaging and measured for 8000 frames. This would be a total imaging time of (exposure time in s * frames recorded) ~ 2 min, which seems to be very short. The authors answered that this exposure time did not include the read-out time per frame and that they imaged for ~ 9 min (exposure time in s * read-out time * frames recorded). But this would lead to the conclusion that the read-out time is larger than the exposure time. In the methods the authors mentioned that they used an air-cooled EMCCD camera (Photometrics Evolve, EVO-512) in the frame transfer mode in combination with the open-source software Micro-Manager. The EMCCD camera used consists of an "image area" and a "storage area". During imaging, the photoelectrons accumulate in the image area and once the

exposure is completed, the entire contents of the image area are rapidly shifted to the shielded storage area. This eliminates dead time between exposures due to continuous exposure by quickly transferring the photoelectrons accumulated in the image area to the storage area while a new exposure begins in the image area. Especially for the EMCCD camera used, the read-out time is in the range of several MHz range, which leads to read-out times of several μ s.

Could the authors please check these parameters mentioned and explain how they calculated 9 min imaging time out of 8000 frames at 15 ms exposure time, especially when the authors used the frame transfer mode?

Reviewer #3

(Remarks to the Author)

The authors responded to all questions, added necessary experiments/data and missing text. Therefore I have no further comments and manuscript can be accepted for publication.

Version 2:

Reviewer comments:

Reviewer #2

(Remarks to the Author)

I would like to thank the authors again for addressing my comments. The authors have addressed all points concerned. The authors have conducted excellent control experiments both with and without UV excitation, demonstrating that after 8000 frames, no significant improvements or differences are observable. In principle, they have also shown that UV excitation can generate faster acquisition times (due to the repopulation of the on-state) with negligible photobleaching. The authors calculated the localization precision using the ThunderSTORM localization software. It would be helpful if they could provide the version number. This software calculates the so-called lateral uncertainty, which considers several parameters in addition to the standard deviation of the PSF, background photons, and pixel size. Generally, this calculation is similar to the calculation according to Mortensen, but not exactly comparable with.

The authors report achieving a localization "uncertainty" of 17 ± 7 nm, which they describe as excellent localization precision. However, based on my interpretation, a very good to excellent localization precision for a classical dSTORM experiment would typically range from 6-10 nm. The reported precision or more correct "uncertainty" is considerably higher than this range and is more comparable to values of good to moderate resolutions (not localization precision). Therefore, I would recommend refraining from using terms like "excellent" and suggest moderating the claims about the significance of these results. It may be more appropriate to use less emphatic language when describing the achieved precision.

Furthermore, the authors have provided an excellent detailed explanation of the composition of the total acquisition time and have appropriately modified this in their methods section. However, it is surprising to see these settings and this approach chosen for high-quality dSTORM imaging.

As the authors have well explained, with the camera settings chosen, the total acquisition time is composed of the sum of the actual exposure time and the readout time. In the selected operating mode, a non-overlap mode is used, which results in the actual recording of blinking events and the readout of this information occurring separately. This leads to a higher dead time (33.7ms readout time) than the actual exposure time of the camera.

Consequently, during recording, the signal is not continuously captured, but rather with an interruption equal to the read-out rate. This means that only less than half of the total acquisition time is used for collecting signal and thus information from the sample. This creates a significant disadvantage.

I truly see the potential, utility, and necessity of this approach. Therefore, I would cautiously recommend publication.

However, I strongly advise modifying the acquisition method and recommend implementing continuous acquisition as a prerequisite, where the readout of information and the recording of the signal (the exposure time) occur in parallel.

This modification would not only significantly strengthen the method but also substantially improve both the quality of the recording and the acquisition time.

Version 3:

Reviewer comments:

Reviewer #2

(Remarks to the Author)

I thank the authors for considering my comments.

In general, the manuscript has improved significantly compared to the first version and after carefully reading the manuscript, I believe that the manuscript can be published as is.

Reply to the Referees

We would like to thank the Referees for their comments. We enclose a point-by-point response and our revised manuscript with the changes highlighted in blue.

Reviewer #1 (Remarks to the Author):

This is a manuscript by a very knowledge group to study several measurements (quantity, morphology) of individual extracellular inflammasome (ASC specks) as biomarker for inflammation in serum, cerebrospinal fluid and brain samples from patients with Parkinson's or Alzheimer's disease.

They first analyzed ASC speck aggregates generated in THP-1 cells and showed that ASC specks form early in the inflammatory response. They then moved into biomarker evaluation of ASC specks numbers, which is the main part of the manuscript. The authors state "...we introduce a new assay that has potential as an ultrasensitive biomarker of inflammation in the blood of AD and PD...":

With sample sizes of CSF (6 PD and 6 HC) and serum (divided in two cohorts: 10 PD and 10 HC as well as 8 PD and 5 HC plus 10 AD vs 10 HC) and soaking brain tissue (9 PD, 5HC and 6 AD) I am not sure if it makes sense to present ROC curves and present this assay as a biomarker of inflammation. I am wondering if the assay is so complicated that larger cohorts are anyway not feasible to analyze? How can this assay be developed for high throughput at all?

➤ The purpose of ROC curves in our paper is to show a potential of our new method to be useful for early diagnosis of neurodegenerative conditions. We appreciate that our ROC curves might be not very smooth with this sample number, and therefore the AUC not very precise, but it is still legitimate and the accuracies indicative. Furthermore the ROC analysis allows for comparison with other biomarkers in development. This is the first paper which publishes this novel method and it is important to show the diagnostic potential.

The method is suitable to scale up in larger patient cohorts. We are currently working on automation of our assay with an aim to run 100 samples per day with a pipetting robot. The current version of the assay can run 40 samples per day using partially-manual pipetting using multi-channel and repeat electronic pipettes. We added a section 'Workflow of SiMPull method' into Materials and Methods on steps of our SiMPull assay to help the reader to follow what we have done.

Importantly in this paper we demonstrate the ultra-sensitivity and reliability of the method, which we think will be of significant interest to the field. We provide a series of validation steps for the method using ASC speck standard samples and quality controls for each biofluid, confirming our ASC speck measurements are analytically reliable with the intra-assay and inter-assay CV < 20% (please see the added section 'Analytical precision of the ASC speck SiMPull assay' for details). The utility of the ASC specks as a biomarker of inflammasome formation in AD/PD is confirmed by significant differences in the number, size and shape of ASC specks between controls and patients with AD and PD.

Authors state several times that they analyzed the samples in EARLY PD and AD, but the MDS-UPDRS numbers and especially also the MMSE mean of 17 in AD is advanced dementia (which can barely be after a mean of 1.6 years of dementia)-something in these cohorts is weird and needs a deeper look.

➤ Thank you. We checked the clinical phenotyping data on each participant and because these patients have been followed longitudinally since the recruitment to a clinic at early-stage AD they had multiple blood collection throughout the clinical course. In our paper, we indeed included the analysis of serum samples at the stage when patients already converted to AD with dementia. The correspondence between MMSE and dementia severity is only approximate. By convention, and in accordance with the UK National Institute for Health and Care Excellence guidelines (NICE; “moderate Alzheimer’s disease: MMSE 10–20”), our patients could be said to have moderate dementia. However, education, literacy, and language deficits can create disproportionately low MMSE for some people with mild dementia and the MMSE does not accurately reflect functional deficits.

Given the reviewers concerns, we have now added a new set of samples from a cohort of 20 people with earlier stage AD, who sought advice at the memory clinic for the first time and had a positive AD CSF biomarker profile (please see the Participants section for details). Overall we have increased our total sample size, and have analysed serum from 20 early-stage AD, 20 later-stage AD (AD with moderate dementia) and 30 cognitively-normal controls (please see Table 2 for the clinical phenotyping data of these participants). Additionally, we analysed the CSF of the same 20 early-stage AD participants and correlated the ASC speck levels between serum and CSF. We observed the levels of ASC specks are well correlated in paired serum and CSF samples from early-stage AD patients (Supplementary Fig. 3F).

In terms of the PD participants, mean duration from diagnosis was only 0.5+/-0.3 years, and the Hoehn and Yahr stages and Addenbrooke’s Cognitive Examination scores (see table 1) confirm that the patients are at an early stage in terms of both motor and cognitive progression. Furthermore the mean MDS-UPDRS-III scores for our cases are very similar to those reported for newly-diagnosed population-representative cohorts (mean of 26.5 in CamPaIGN cohort and mean of 31.6 in PICNICS cohort – see Evans JR, Cummins G, Breen DP, Foltynie T, Mason SL, Brayne CE, Williams-Gray CH, Barker RA. Comparative epidemiology of incident Parkinson's disease in Cambridgeshire, UK. *Journal of neurology, neurosurgery, and psychiatry*. 2016;87(9):1034-6.)

The authors next performed some control experiments indicating aggregate specificity and assay stability including conformation stability. Here, I think depletion experiments are lacking showing the depletion of the assayed signal.

Thank you for the advice. We’ve now added the immunodepletion experiment for n = 2 PD serum samples (Supplementary Fig. 3G) to the revised paper which demonstrate the depletion of the assayed signal in the serum sample after immuno-depletion of ASC protein using our previously-published IP protocol for serum (Lobanova, E. et al. Imaging protein aggregates in the serum and cerebrospinal fluid in Parkinson’s disease. *Brain* 145, 632–643 (2022)).

For more specificity the authors added to the number of ASC specks (and later also morphologic aspects) the numbers of aggregates of beta-amyloid, tau and alpha-synuclein for calculating ratios to the ASC specks as “composite biomarker”. Here it would be important to show the specific markers alone. Also established SAA methods would be good to add. I am also lacking inflammatory marker protein(s) here. Authors do state in the method section, that the conditioned cell media was also assayed for IL-1beta by ELISA to confirm inflammasome activation. Such a quantification should also be carried out in the biofluids to correlate with the ASC Speck numbers and morphology.

Overall, I think this is an interesting assay and way to visualize and analyze inflammasomes.

➤ Thank you for the suggestions. For those subjects for whom we had sufficient time-matched serum samples available, we measured a panel of inflammatory cytokines (IL-1 β , IFN- γ , IL-2, IL-4, IL-6, IL-10, IL-12p70, IL-17A, TNF- α). We were able to do this in our cohort of 20 AD dementia serum samples and 10 HCs, as well as 7 PD and 11 HCs. Measurements were performed using the Mesoscale Discovery (MSD) S-plex assay (please see Supplementary Figure 7). Please see the results of correlation with the ASC speck measurements in Supplementary Figure 7 and the added text on page 6. We found that ASC specks positively correlated with the cytokine IL-1 β levels in PD and AD serum samples as well as healthy controls. The cytokines IL-17A and IL-10 also positively correlated with ASC speck in PD patient's serum whereas the IL-2 inversely and IL-10 positively correlated with the ASC speck in AD dementia serum. CRP positively correlated with the ASC speck in HC but not AD dementia serum. As for the diagnostic value, serum ASC speck shows a better dynamic range (~2-4 fold increase in AD and PD serum compared to controls) than IL-1 β (1.3-1.4 increase in AD and PD serum), and greater sensitivity to differentiate PD/AD from control serum (ASC speck: AUC = 89% AD vs HC serum, AUC = 87% PD vs HC serum compared to IL-1 β : AUC = 66% AD vs HC serum, AUC = 73% PD vs HC serum).

The individual single-molecule markers of beta-amyloid and p-tau used for calculating ratios for AD and HC serum groups are now shown in Supplementary Fig. 3A and B. The single beta-amyloid and alpha-synuclein are added to Supplementary Fig. 3H-K.

I am not convinced about the potential for such an assay as a robust biomarker for larger cohorts, rather could this assay be utilized to detect inflammation response very early in the disease. Existing cohorts of prodromal subjects and with different disease stages would be necessary here.

➤ We are developing the automation of our assay with which we will be able to run 100 samples per day. Regarding the use of our assay to detect inflammation in 'preclinical' disease stage, we now included a cohort of 20 early-stage AD vs 20 HC paired serum and CSF samples from the same individuals subjects. In these individuals, a pre-clinical AD diagnosis was made according to the 2018 NIA-AA research framework (<https://doi.org/10.1016/j.jalz.2018.02.018>) in which the participants seeking advice in a memory clinic for the first time and had abnormal AD biomarkers (positive CSF A β 42/40 ratio and CSF p-tau profile). AD CSF biomarker profile was defined by a CSF A β 42/p-tau181 ratio < 10.25 (as measured by Lumipulse G600II, Fujirebio) (Table 2). Using our ASC SimPull assay, we were able to detect an increased quantity of ASC specks in both serum and CSF from early-stage (pre-clinical) AD patients compared to controls (Fig. 2D, p = 0.0046 for n = 20 early AD vs 20 HC serum; Fig. 2F, p = 0.0005 in CSF).

Authors state at the end of the discussion that their assay might be useful as a biomarker to detect target engagement of inflammasome inhibitors in clinical trials-I think this cannot be stated with the data shown as neither performed the authors inflammation inhibition nor enough data to claim this as a biomarker.

➤ We were only stating the method might have potential for this, not that we have in this paper proven this potential. This potential use will be of interest and high relevance, given that there are several trials of inflammasome inhibitors underway or in set-up. However, we have removed this statement from the discussion and allow readers to consider the potential as they wish.

Minor:

some abbreviations of the figures are not included in the legends.

➤ Resolved, thanks.

How was the neuropathological diagnosis of case PD8 been done if Lewy Body Braak staging is missing? Knowing of co-aggregation and co-pathology all staging should be done for all brains.

The post mortem neuropathological report confirmed Lewy pathology in the medulla, pons and midbrain, thus confirming the diagnosis of Parkinson's disease, but the staging information is missing for this case.

Reviewer #2 (Remarks to the Author):

The authors present a novel smart and easy single-molecule pull-down assay (combined with dSTORM) characterizing specific crucial parameters of individual ASC specks in serum, CSF and brain from PD and AD patients. The authors achieved greater biomarker performance and accuracy by combining the quantity of ACS specks with other biomarkers of protein aggregate diseases. This novel approach has the potential to distinguish healthy people from people at early stages of aggregate diseases (PD & AD) as well as to potentially characterize specific therapeutic treatments at the single-molecule level in clinical trials.

The manuscript is well written and most of the illustrations are clear to understand. Overall, this work is a valuable contribution to the field. However, I would have some crucial questions which need to be addressed prior publishing the paper in Nature Communications.

Major points:

1. Regarding the coverslip preparation, the authors refer to reference #39 which can not be found online (maybe also in review or recently submitted? [not stated]) (see also minor point #3). Furthermore, the authors did plasma cleaning and afterwards a hydrophobic self-assembling monolayer approach using Rain-X. Afterwards, this prepared surface is treated with „biotinylated NeutrAvidin“ but where does the biotin binds to? It is clear, that the authors use NeutrAvidin to immobilize the biotinylated capture antibody but it is not clear why the NeutrAvidin must be biotinylated? (see Fig. 1A). Or is this an error in the illustration? There is also no protein on the surface or at least stated in the manuscript that will bind to the biotinylated NeutrAvidin to immobilize it. The authors also blocked the treated surface with BSA, but wouldn't it be obvious to only use a mixture of BSA and biotinylated BSA or even only biotinylated BSA to assure the binding of NeutrAvidin, building up the SiM Pull assay?

➤ Thanks for your important comments. Our “surface paper” is under peer-review in the ACS Nano journal but we now attached the manuscript for support. The surface was treated with label-free NeutrAvidin and we now corrected the Fig. 1A. After adding the biotinylated capture antibody, we applied polymer f127 and BSA for blocking the surface (filling the gaps where there is no biotinylated antibody) before adding the sample which helped us to reduce non-specific binding of ASC specks to the surface.

2. The authors did certain control measurements including SiMPull assay without capture antibody as well as SiMPull with capture antibody but with a non-target Alexa647 IgG which sounds sensible. But to be sure, concerning unspecific binding, I would prefer to also test the SiMPull assay without the capture antibody but with the correct target Alexa647 IgG in buffer only, THC-1 cell lysate and THC-1 cell media to validate unspecific binding of the detecting antibody (see also minor point #11).

➤ Thanks for your suggestions. The media from the non-stimulated THP1 cells ('no stimulation media') with the right capture and correct target Alexa647 antibody worked as our buffer control in Fig 1E cells. We also included 'no capture' control for which we used the stimulated THP-1 cell lysate and THP-1 cell media without the capture antibody and correct target Alexa647 antibody to validate no unspecific binding of the detection antibody (Fig 1B).

3. In Fig. 1B+E the authors showed box plots of the SiMPull assays measured. „The dots in B and E represent independent replicates n=3“. But for this sample size (n=3) it is hard to generate adequate statistics. For these crucial experiments of the SiMPull assay, I prefer to increase the sample size to at least n = 5 or even better n = 7-10 independent measurements.

➤ In this experiment, each independent replicate ('dots in B and F') is an average of 12 different fields of view (images). We now included this description to the Figure 1 caption.

4. The dSTORM measurements were done with an exposure time of 15 ms (66.67 Hz) for 2 min (8000 frames), with a continuous UV illumination (405 nm). For high quality dSTORM measurements an appropriate switching behaviour of the fluorophores used has to be reached. The individual fluorophores should show long non-fluorescent „off-states“ (several seconds) and short fluorescent „on-states“ (several milliseconds). At the beginning of an experiment, most fluorophores should be transferred into the off-state by irradiating the whole sample in the FOV with the appropriate laser wavelength and power (if measurements are done in the TIRF mode, ensure that the fluorophores show an appropriate switching behaviour, which can easily be achieved by changing the illumination mode to EPI for a short period of time and then go back to the TIRF mode for measurement). This „off-state“ can be elongated using oxygen scavenger systems like the system the authors used. This ensures, that the triplet state as well as the „off-state“ is not quenched by molecular oxygen. Adequate off-state times reduce the risk of overlapping signals (overlapping point spread functions (PSF)) from neighboring fluorophores in the diffraction-limited space, minimizing the risk of (blinking) artifacts and boosting the quality of the localization data and high resolved images. The on-state of the fluorophores used (Alexa647) can also be repopulated by UV light. This could also increase the probability of multiple fluorophores in the on-state within the diffraction-limited space. Which could lead to high amounts of artifacts, false localizations and bad data quality. The quality of the images shown varies greatly and the images reflect the localization data. Maybe the authors can also upload a dSTORM movie to have a look at the switching behaviour. As stated previously, the authors measured for 2 min (8000 fr), which seems to be a very short measuring time. Maybe because of that, the authors used continuous UV illumination to detect a high amount of emitters within this short measuring time. But this could adversely affect the quality of the measured dSTORM localization data. But this is a crucial point for the whole paper and method. To improve the quality of the images as well as the localization data, I would recommend that the authors also do some measurements of the SiMPull assay for the THP-1 cell lysate and media without continuous UV illumination (like for Fig. 1 B and E) for 5 – 10 min (afterwards the movie can also be shortened to 2 min if there is no further positive effect on the localization data quality after to 2min) to minimize the risk of bad blinking behaviour and overlapping PSF (within this short measuring time) improving the quality of the data.

➤ Our exposure time of 15 ms didn't include read out time per frame. Therefore, our dSTORM measurements were done for 8000 frames which took about 9 min. As you suggested, we checked whether the size and shape of aggregates change when increasing the number of frames to 16k. We did this using PD serum and observed no significant difference (within a 95% confidence interval) in the size and shape distributions of ASC aggregates in serum when imaged with 16k frames instead of 8k frames (please see the Supplementary Fig.6).

Minor points:

1) The authors are doing direct stochastic optical reconstruction microscopy (dSTORM) but cite a publication about „STORM“ using FRET via Cyanine3 and Cyanine5 to achieve blinking of Cy5. So I would recommend to cite a dSTORM paper where Cyanine dyes are directly photochemically switched between on- and off-state. (This refers to reference #13: Rust, M. J., Bates, M. & Zhuang, X. Sub-diffraction-limit imaging by stochastic optical reconstruction microscopy (STORM). Nat Methods 3, 793–795 (2006))

➤ We now added the right citation #13.

2) It is common practice to italicize the letter d in dSTORM, so I would recommend to change all d in dSTORM to italicized letters. And I would also recommend to italicize the word direct when writing out dSTORM (page 3). Furthermore, the authors often state STORM instead of dSTORM, so please change STORM to dSTORM.

➤ Corrected, thanks.

3) Concerning some references, it is not possible to find the reference #29 (Zhang, Y. P. et al. Imaging protein aggregates in Parkinson’s disease serum using aptamer-assisted single-molecule pull-down. Chem Sci (2023)) as well as #39 (Zhang, Y. P. et al. A simplified imaging surface for versatile and high-throughput single-molecule studies. Anal Chem (2023)). The authors have stated the journal but without any additional specific numbers like edition or pages. Are these papers already published, in a review process or even recently submitted and the authors forgot to mention?

➤ Our ‘surface paper’ is under peer-review in the ACS Nano but we now attached the manuscript for support.

4) Beside the previous remark, could the authors also complete reference #6 (King, E. et al. Inflammation in Mild Cognitive Impairment due to Parkinson’s disease, Lewy Body disease and Alzheimer’s disease) „... Int J Geriatr Psychiatry. 2019 Aug;34(8):1244-1250. doi: 10.1002/gps.5124. Epub 2019 May 16. PMID: 30993722“ is missing

➤ Corrected, thanks

5) On page 4 it is stated: „The SiMPull assay is capable of detecting ASC specks in both the NLRP3 inflammasome-activated THP-1 cell lysates (LPS-primed nigericin-activated) and those secreted into the media compared to untreated negative controls (Fig. 1C)“. The authors refer to Fig. 1C, but doesn’t refer this sentence to Fig. 1B/E or or Fig. 1C/D or at least to Fig. 1B-E and not only to Fig. 1C?

➤ Corrected, thanks

6) Regarding Fig. 1A: Could the authors make the caption a little bit more detailed, for example what the yellow dots are or belong to? So that at the end it is easier to follow only by looking at the illustration of the SiMPull assay.

➤ For simplicity we now removed the yellow dots but they meant non-ASC aggregates which the ASC antibody should not bind

7) In Fig. 1A please correct „biotinylated ASC (AL117) antibody“ to antibody.

➤ Corrected, thanks

8) In Fig.1, B and E is missing.

➤ Corrected, thanks

9) Concerning Figure 1: For clearer illustration I would recommend rearranging the order of the images and graphs.

a. Top row: A) ASC SiMPull (left), B + C) THP-1cell lysates and cell media (dSTORM-pictures) (right)

b. Middle row: D) THP1-cell lysates box plot (left), E) media box plot (right) if possible with the same y-axis scale to directly see the differences

c. Lower row: F and G (aggregate area and circularity)

➤ Rearranged as suggested, thanks

10) I would also recommend to change the way of illustration in Fig. 1 F and G, because it is very confusing to correctly assign the geometric shapes. I would prefer to use clearly distinguishable shapes or even different colours. (The design of the other box plots like Figure 2 etc. is way easier since there is only a distinction between two groups HC and PD or AD)

➤ Changed as suggested, thanks

11) Which degree of labeling (DOL) did the authors use for the target Alexa647 antibody? Because it is also well known, that higher DOL ($>>4$) could also lead to a high amount of unspecific binding of the antibodies.

➤ DOL for our Alexa647 AL177 antibody was 4-5.

12) In the method section under „ASC speck SimPull assay“, in the second sentence the authors mentioned the word „twice“ twice, so please delete one twice („...washed twice with PBS twice...“).

➤ Corrected, thanks

Reviewer #3 (Remarks to the Author):

In the article by Lobanova et al, the authors describe the use of ASC inflammasome specks as a sensitive fluid biomarker of neuroinflammatory disease severity in Parkinson's and Alzheimer's disease. The authors claim that the increased quantity and abnormal morphology of ASC specks in the blood of patients makes them a candidate biomarker of inflammation for Parkinson's and Alzheimer's diseases. When combined with other measurements the diagnostic accuracy can be further improved.

The article is clearly written and the study was well designed and implemented. The conclusions are supported by the results to a large extent. However some issues need

Since the study is based on measuring inflammation as a biomarker of PD/AD it should be considered to include some of the more common markers of inflammation in the analysis (C-reactive protein, IL-

1 β , IL-6, INF- γ , MCP-1, TNF- α , erythrocyte sedimentation rate and plasma viscosity) to see how they correlate with ASC specks. And to see if ASC specks present the added value for the diagnosis.

- For those subjects for whom we had sufficient time-matched serum samples available, we additionally measured the concentrations of IL-1 β , IFN- γ , IL-2, IL-4, IL-6, IL-10, IL-12p70, IL-17A, TNF- α cytokines and C-reactive protein (CRP) levels using Mesoscale Discovery assays. This was done in 20 AD with dementia and 10 HC serum samples. In our PD cohort, we also measured the same cytokine panel in 7 PD and 11 HC serum samples. Please see the results of correlation in Supplementary Figure 7 and the added text in page 6. We found that ASC specks positively correlate with the cytokine IL-1 β levels in PD, AD serum samples as well as healthy controls. The cytokines IL-17A and IL-10 also positively correlate with ASC speck measurements in PD serum whereas the IL-2 inversely and IL-10 positively correlate with the ASC speck in AD dementia serum. CRP only positively correlates with the ASC speck in HC but not AD dementia serum. As for the diagnostic value, serum ASC speck shows a better dynamic range (~2-4 fold increase in AD and PD serum compared to controls) than IL-1 β (1.3-1.4 increase in AD and PD serum), and greater sensitivity to differentiate PD/AD from control serum (ASC speck: AUC = 89% AD vs HC serum, AUC = 87% PD vs HC serum compared to IL-1 β : AUC = 66% AD vs HC serum, AUC = 73% PD vs HC serum).

Also for CSF and plasma samples, were patients free of any disease that can cause even minor inflammation? Was white blood cell count performed on CSF and blood?

- Potential participants were excluded if they had had a current or recent infection, were taking immunomodulatory/anti-inflammatory medications or had vaccinations within the last 3 weeks; and we excluded those with a history of known systemic inflammatory disease (e.g. rheumatoid, Crohn's, systemic lupus erythematosus etc). We cannot exclude minor undeclared or asymptomatic inflammatory lesions that can occur in older adults, but refer also to the age-matched control group. White cell counts are unfortunately not available for the CSF and blood samples used in this study.

Since the study is based on inflammation markers, shouldn't the AD case with Burkitt's Lymphoma and pneumonia be excluded from the analysis?

- As suggested, we have excluded these cases from the paper.

In addition, when it is not stated, what was the death cause? Was it always natural, or some other death causes were present?

- Cause of death can be very difficult to quantify particularly when patients die in the community, and reliable information is unfortunately not available for all post mortem cases. Our patients did not die by violence or suicide, so they may be defined as "natural causes": but the agonal mediators are often not known or are pathophysiologically ill-specified for people with dementia or advanced age.

We have included what is available from post mortem reports. We feel it is appropriate to include all available post mortem tissue in the study in spite of incomplete clinical data, given that such tissue is a highly limited and valuable resource.

The CSF collection site is not specified. It is well known that protein gradients exist in CSF and differences in lumbar puncture sites, and even patient height, will affect the results of any assay. To omit the problem the measured values should be normalized to for example the total protein level.

- For all our participants, CSF was collected at the same site - L3/4 level. Unfortunately we have now used all of our aliquots of CSF samples included in this study and so have not measured total protein levels, but looking into our ASC speck assay data we were able to measure a robust increase in the number of ASC specks in both early PD and AD CSF compared to controls. Additionally, we now added a new cohort of 20 early AD and 20 HC paired CSF and serum samples which allowed us to validate our observations, and we also observed a strong correlation in the ASC speck levels between CSF and serum in matched samples (please see Supplementary Fig. 3F for detail). Furthermore, our method is looking at ratios of aggregates (ASC/A β for example) which allows us to improve the diagnostic accuracy in AD/PD serum and mitigates the problem of normalisation to the total protein level.

Another aspect is if the CSF was tested for contamination with blood. Some measurements will be strongly affected by even a small amount of blood contamination.

Any visibly contaminated CSF samples were not used for this study. During the collection procedure, the first few drops of CSF were routinely discarded to minimise the risk of blood contamination.

Finally, the discussion lacks the section about the limitations of the study. For example, authors claim that their method can be used as a biomarker, however how easy will it be to implement it in routine clinics? How long the test will take? What will be the cost of analysing the sample?

- We now added a section ('Workflow of SiMPull method' in the Methods section) describing our assay steps and also extended Fig. 1A. Running a 40-well plate with all measurements of quantity, size and shape of ASC aggregates takes 14h of which 10 hours are done remotely with automated image acquisition. The instrument costs ~£70K and each 40-well plate has minimal cost because we use very small amount of antibodies (1 ul per antibody per plate). We are also currently working on our assay automation for which we programmed a pipette robot (Analytik Jena CyBio Felix robot) to automatically prepare the surface coating, perform washing steps, add the surface capture antibodies, add the samples and then add the imaging antibodies.

As you suggested, we added a limitations of study section at the end of the paper.

Minor issues

No post-mortem interval is specified for AD patients.

- We now removed our post-mortem AD brain cases from the study because as you suggested two of them had cancer or pneumonia at the time of death.

Figure 1 – Only A, F and G captions are visible

- Corrected, thanks

Scale bars on the figures should start from 0

- Changed, thanks

Reply to the Referees

We would like to thank the Referees for their comments. Addressing the Referees' comments, we prepared the Rebuttal where we address each Referee's point one-by-one. Please find our Response below.

REVIEWER COMMENTS

Reviewer #1 (Remarks to the Author):

The manuscript has definitively improved with the new independent samples, the additional inflammatory values and methodological aspects. I still see quite some incongruency in the cohorts (quite high UPDRS numbers of short duration etc.) and despite the comments by one other reviewer there are a significant number of references that are not complete.

- Thanks for your comments. We now completed all of our references except for one reference (Zhang, Y. P. et al. An improved imaging surface for quantitative single-molecule microscopy <https://doi.org/10.17863/CAM.109785>) which have been just accepted for publications in ACS Applied Materials & Interfaces journal and will be able to complete this reference once it is published on the journal web.

For both of our PD cohorts, disease duration was short as we aimed to assess the new biomarkers in early stage disease. We used the Hoehn and Yahr (HY) classification to determine the stage of the disease. All of our patients were in HY stages 1-2 except for 1 patient who was HY stage 3. Whilst there is some unavoidable heterogeneity in MDS-UPDRS scores across the participants, the range and mean scores are consistent with what we would expect for early-stage PD based on our experience of assessing PD patients in our research clinic for many years. The UPDRS-III scores of the included participants are very similar to those observed in our large newly-diagnosed population-representative cohorts (mean of 26.5 in CamPaIGN cohort and mean of 31.6 in PICNICS cohort – see Evans JR, Cummins G, Breen DP, Foltynie T, Mason SL, Brayne CE, Williams-Gray CH, Barker RA. Comparative epidemiology of incident Parkinson's disease in Cambridgeshire, UK. *Journal of neurology, neurosurgery, and psychiatry*. 2016;87(9):1034-6.) Other authors also concur that scores around this level are consistent with mild/early disease. Martinez-Martin et al suggest that MDS-UPDRS Part III score < 32 is indicative of mild PD (Martínez-Martín P. et al. Parkinson's disease severity levels and MDS-Unified Parkinson's Disease Rating Scale. *Parkinsonism Relat Disord*. 2015 Jan;21(1):50-4). Skorvanek et al. reported MDS-UPDRS Part III scores of 29.8 ± 15.5 [0–97] for their studied cohorts within 5 years of PD diagnosis which are similar to our values (UPDRS Part III = 29.0 ± 15.2 [19–63]) (Skorvanek M. et al. Differences in MDS-UPDRS Scores Based on Hoehn and Yahr Stage and Disease Duration. *Mov Disord Clin Pract*. 2017 Mar 11;4(4):536-544.)

Reviewer #2 (Remarks to the Author):

I would like to thank the authors very much for addressing and revising most of the points and comments mentioned.

However, there are some crucial points that have still not been addressed.

- 1) The authors performed dSTORM (direct Stochastic Optical Reconstruction Microscopy) measurements, but they are still referring to STORM (Stochastic Optical Reconstruction Microscopy),

which has some differences from dSTORM. Please modify the respective reference (#13) accordingly and cite a dSTORM paper (e.g., M. Heilemann, S. van de Linde, or M. Sauer), for example M. Heilemann, S. van de Linde, M. Schüttpehl, R. Kasper, B. Seefeldt, A. Mukherjee, P. Tinnefeld, and M. Sauer, "Subdiffraction-resolution fluorescence imaging with conventional fluorescent probes," *Angewandte Chemie International Edition*, vol. 47, no. 33, pp. 6172-6176, 2008"

➤ Corrected, thanks.

2) The authors are still writing STORM instead of dSTORM. Please correct it to dSTORM in the main text as well as in the supplementary information.

a. main text

i. In line 617 of the revised manuscript

ii. In line 687

iii. Figure caption 5. Line 972

b. Supplementary information

i. Figure caption 4, line 2

ii. Figure caption 5, line 1

3) Please also italicize the letter d in dSTORM in the entire manuscript

➤ Corrected, thanks.

4) The authors have not really answered the major point 4 regarding the performance of the dSTORM measurements.

a. Why do the authors use continuous UV illumination? On the one hand, the on-state of the fluorophores used (Alexa647) can be repopulated by UV light. This could increase the probability of multiple fluorophores in the on-state within the diffraction-limited space, especially if the labeling density of the target structure is high. This could lead to high number of artifacts, false localizations and bad data quality. The quality of the images shown varies greatly and the images reflect the localization data. On the other hand, continuous UV illumination leads to a higher risk of faster photobleaching of the dyes, resulting in a rapid loss of emitters and information, especially with a high irradiation power of 10 mW like the authors stated in the methods. To minimize the aspects stated no or pulsed UV illumination would be better. Could the authors also do some simple control measurements without UV illumination? Since I recommended to do some measurements of the SiMPull assay without continuous UV illumination to minimize the risk of bad blinking behavior and overlapping PSF improving the quality of the data.

➤ Thank you for your important comment. The UV laser in our current set-up can't function in pulsed mode and therefore we used continuous UV illumination at power 10mW (at the source) for active photoswitching. As suggested, we have performed a control experiment with and without UV laser illumination measured on an AD CSF sample. Then, we compared the number of localisations per frame (in ms) measured with and without the UV laser (please see the Supplementary Fig 10 E) and didn't observe a significant photobleaching with the UV laser. However, the total number of localisations without UV illumination was reduced ~2-fold (Supplementary Fig 10E) and not all

diffraction-limited aggregates were present in a super-resolved image resulting in low aggregate number per FOV (Supplementary Fig 10A,B). Average number of localizations per ASC aggregate with the UV laser was 58 compared to 64 without the UV (see Supplementary Fig 10F). The average localisation precision in our dSTORM images was 17 ± 7 nm (Supplementary Fig 10G), with > 95% of localisations having excellent precision (low uncertainty) < 30 nm. This indicated a low risk of detecting multiple emitters in the same frame (overlapping point spread functions).

We have also checked that the use of continuous UV laser illumination didn't result in photobleaching of all fluorophores after 8k frames, as evidenced by sufficient number of localisations we detected from 8k-16k frames (Supplementary Fig 6E). To investigate further how the size and shape of aggregates change when increasing the number of frames, we included the statistical data where we compared the interquartile ranges (IQR) of the size and shape distributions imaged with 2k, 5k, 8k, 11k and 14k frames (please see Supplementary Fig.9) and found that the morphological information on ASC aggregates started to have a stable distribution with 8k frames (area IQR = $0.015 \mu\text{m}^2$ and circularity IQR = 0.287, Supplementary Fig 9). Taken together, this confirms that the morphology measurements of ASC aggregates saturate at 8k frames due to sufficient sampling (limited by the number of fluorophores on each aggregate).

b. The authors mentioned that they used 15 ms exposure time for imaging and measured for 8000 frames. This would be a total imaging time of (exposure time in s * frames recorded) ~ 2 min, which seems to be very short. The authors answered that this exposure time did not include the read-out time per frame and that they imaged for ~ 9 min (exposure time in s * read-out time * frames recorded). But this would lead to the conclusion that the read-out time is larger than the exposure time. In the methods the authors mentioned that they used an air-cooled EMCCD camera (Photometrics Evolve, EVO-512) in the frame transfer mode in combination with the open-source software Micro-Manager. The EMCCD camera used consists of an "image area" and a "storage area". During imaging, the photoelectrons accumulate in the image area and once the exposure is completed, the entire contents of the image area are rapidly shifted to the shielded storage area. This eliminates dead time between exposures due to continuous exposure by quickly transferring the photoelectrons accumulated in the image area to the storage area while a new exposure begins in the image area. Especially for the EMCCD camera used, the read-out time is in the range of several MHz range, which leads to read-out times of several μs .

Could the authors please check these parameters mentioned and explain how they calculated 9 min imaging time out of 8000 frames at 15 ms exposure time, especially when the authors used the frame transfer mode?

- Thank you for your comment. We checked throughout the imaging parameters we used and for dSTORM we acquired 8000 frames at 30ms camera exposure time (not 15 ms) and 10MHz 16bit read-out time (equivalent to a frame rate of 33.7 ms per frame). The Evolve 512 camera was operated in non-overlap mode (non-simultaneous exposure-readout) with the parameters: clearing mode = pre-exposure and clocking mode = frame transfer as recommended by the manufacturer. In non-overlap mode, exposure and readout are carried out in non-overlapped fashion and as a result each frame in the sequence is precisely exposed for the time specified (i.e. 30 ms). Taking together, the total time to take 8000 frames was ~8.5 min ($8000 \times 30 \text{ ms} + 8000 \times 33.7 \text{ ms}$) + 0.5min to save the image on computer disc after acquisition. In the Imaging section, we made the required changes to the detailed imaging parameters used. The use of 8000 frames appears to be sufficient (please see Supplementary

Fig.6 and now added Fig.9) to measure the size and shape differences between AD/PD and controls and in this paper we want to show a potential of our method to be scalable for diagnostics with the aim to run hundreds of patient's samples per day.

Reply to the Referees

We would like to thank the Referee for their comments and suggestions. Addressing the Referee's comments, we prepared the Rebuttal where we address each Referee's point one-by-one. Please find our Response below and changes that are highlighted in the manuscript and SI.

REVIEWER COMMENTS

Reviewer #2 (Remarks to the Author):

I would like to thank the authors again for addressing my comments. The authors have addressed all points concerned. The authors have conducted excellent control experiments both with and without UV excitation, demonstrating that after 8000 frames, no significant improvements or differences are observable. In principle, they have also shown that UV excitation can generate faster acquisition times (due to the repopulation of the on-state) with negligible photobleaching.

The authors calculated the localization precision using the ThunderSTORM localization software. It would be helpful if they could provide the version number. This software calculates the so-called lateral uncertainty, which considers several parameters in addition to the standard deviation of the PSF, background photons, and pixel size. Generally, this calculation is similar to the calculation according to Mortensen, but not exactly comparable with.

➤ We used the version 1.3 and now mentioned this in the manuscript.

The authors report achieving a localization "uncertainty" of 17 ± 7 nm, which they describe as excellent localization precision. However, based on my interpretation, a very good to excellent localization precision for a classical dSTORM experiment would typically range from 6-10 nm. The reported precision or more correct "uncertainty" is considerably higher than this range and is more comparable to values of good to moderate resolutions (not localization precision).

Therefore, I would recommend refraining from using terms like "excellent" and suggest moderating the claims about the significance of these results. It may be more appropriate to use less emphatic language when describing the achieved precision.

➤ Thanks for your comments. As you suggested, we lowered down our claims and now noted in the manuscript our resolution to be moderate.

Furthermore, the authors have provided an excellent detailed explanation of the composition of the total acquisition time and have appropriately modified this in their methods section. However, it is surprising to see these settings and this approach chosen for high-quality dSTORM imaging.

As the authors have well explained, with the camera settings chosen, the total acquisition time is composed of the sum of the actual exposure time and the readout time. In the selected operating mode, a non-overlap mode is used, which results in the actual recording of blinking events and the readout of this information occurring separately. This leads to a higher dead time (33.7ms readout time) than the actual exposure time of the camera.

Consequently, during recording, the signal is not continuously captured, but rather with an interruption equal to the read-out rate. This means that only less than half of the total acquisition time is used for collecting signal and thus information from the sample. This creates a significant disadvantage.

I truly see the potential, utility, and necessity of this approach. Therefore, I would cautiously recommend publication. However, I strongly advise modifying the acquisition method and recommend

implementing continuous acquisition as a prerequisite, where the readout of information and the recording of the signal (the exposure time) occur in parallel.

This modification would not only significantly strengthen the method but also substantially improve both the quality of the recording and the acquisition time.

- Thanks for your important suggestions. As suggested, we first performed control *d*STORM experiment where we compared overlap vs non-overlap mode. We found that both camera modes gave us similar results for the size and shape distributions of ASC specks in the CSF samples (within 95% confidence interval) and similar average lateral uncertainty of 13 ± 5 nm (compared to 13 ± 6 for the non-overlap mode) (Supplementary Fig.11). However, since the overlap mode significantly speeds up the acquisition time which is a big advantage for high-throughput imaging and diagnostic applications, we added new data on *d*STORM imaging of ASC specks in 14 early-stage AD and 14 age/sex-matched control human CSF from our study cohort. With the faster imaging time, this allowed us to observe robust differences in the size and shape of ASC specks between disease and control groups which become smaller in disease similar to what we observed for AD/PD serum and soaked brain (please see a new Figure 7 in the manuscript). We also added a new section to the paper “Higher-throughput morphology analysis of ASC specks in human biofluids using faster *d*STORM” to highlight how the method can be improved using the overlap camera mode.